# Environmental variables drive plant species composition and distribution in the moist temperate forests of Northwestern Himalaya, Pakistan

**Inayat Ur Rahman**[1,2]*, **Robbie E. Hart**[2]*, **Farhana Ijaz**[1], **Aftab Afzal**[1], **Zafar Iqbal**[1], **Eduardo S. Calixto**[3,4], **Elsayed Fathi Abd_Allah**[5], **Abdulaziz A. Alqarawi**[5], **Abeer Hashem**[6], **Al-Bandari Fahad Al-Arjani**[6], **Rukhsana Kausar**[7], **Shiekh Marifatul Haq**[8]

1 Department of Botany, Hazara University, Mansehra, Khyber Pakhtunkhwa, Pakistan, 2 William L. Brown Center, Missouri Botanical Garden, St. Louis, MO, United States of America, 3 Entomology and Nematology Department, University of Florida, Gainesville, FL, United States of America, 4 Department of Biology, University of Missouri St. Louis (UMSL), Saint Louis, MO, United States of America, 5 Department of Plant Production, College of Food and Agriculture Science, King Saud University, Riyadh, Saudi Arabia, 6 Botany and Microbiology Department, College of Science, King Saud University, Riyadh, Saudi Arabia, 7 Department of Environmental Sciences, International Islamic University, Islamabad, Pakistan, 8 Department of Botany, University of Kashmir, Hazratbal, Srinagar, Jammu & Kashmir, India

☯ These authors contributed equally to this work.
* hajibotanist@outlook.com (IUR); robbie.hart@mobot.org (REH)

**Data Availability Statement:** All relevant data are within the paper.

**Funding:** This study was supported by the Higher Education Commission (HEC), Pakistan, through a

## Abstract

By assessing plant species composition and distribution in biodiversity hotspots influenced by environmental gradients, we greatly advance our understanding of the local plant community and how environmental factors are affecting these communities. This is a proxy for determining how climate change influences plant communities in mountainous regions ("space-for-time" substitution). We evaluated plant species composition and distribution, and how and which environmental variables drive the plant communities in moist temperate zone of Manoor valley of Northwestern Himalaya, Pakistan. During four consecutive years (2015–2018), we sampled 30 sampling sites, measuring 21 environmental variables, and recording all plant species present in an altitudinal variable range of 1932–3168 m.a.s.l. We used different multivariate analyses to identify potential plant communities, and to evaluate the relative importance of each environmental variable in the species composition and distribution. Finally, we also evaluated diversity patterns, by comparing diversity indices and beta diversity processes. We found that (i) the moist temperate zone in this region can be divided in four different major plant communities; (ii) each plant community has a specific set of environmental drivers; (iii) there is a significant variation in plant species composition between communities, in which six species contributed most to the plant composition dissimilarity; (iv) there is a significant difference of the four diversity indices between communities; and (v) community structure is twice more influenced by the spatial turnover of species than by the species loss. Overall, we showed that altitudinal gradients offer an important range of different environmental variables, highlighting the existence of micro-climates that drive the structure and composition of plant species in each micro-region. Each plant community

scholarship awarded to IUR as an IRSIP research fellow to conduct research work at the Missouri Botanical Garden, Saint Louis, MO, USA. The authors would like to extend their sincere appreciation to the researchers supporting (Project Number RSP-2021/134), King Saud University, Riyadh, Saudi Arabia.

**Competing interests:** The authors have declared that no competing interests exist.

along the altitudinal gradient is influenced by a set of environmental variables, which lead to the presence of indicator species in each micro-region.

## Introduction

Mountains are the most remarkable landforms on the earth, representing different vegetation zones based on environmental variations [1]. They offer habitat heterogeneity based on micro-environmental variation along the altitudinal gradient [2, 3], where environmental variables (including direct and indirect effects of abiotic and biotic effects) are important factors in determining altitudinal zone boundaries [4, 5]. It is well recognized that altitude is a dynamic gradient along which several environmental variables [6] and species diversity [7, 8] change concomitantly. Within this focus, plant biodiversity is strongly influenced by different environmental variables [9], and certain species can survive on the brink of extinction in high mountains across the world [10–12].

Many mountains across the globe are important hotspots of biodiversity [13–16], with roughly half of all plant species flourishing in hotspot areas [17, 18]. However, despite this high endemism of species greatly influenced by various environmental gradients, such as edaphic, climatic and physiographic variables [19], these areas suffer a major impact from climate change [20, 21]. Plant species present in these gradients have great adaptive power [22, 23]; however, the speed with which climate change is advancing might not be sufficiently achieved by plant species adaptation in these areas, leading to a strong impact on the biodiversity of these areas [24–26], ultimately leading to variations in the community structure [27].

In the face of the current climate change and considering the importance of phytosociological studies for the understanding of biodiversity and species distribution, the Himalayas represent a fundamental piece for these studies. This region is facing a marked increase in temperature [28], which is three times greater than the global average [29]. This unprecedented rise in temperature, and modification of various environmental variables as well, may lead to shifts in species composition [24–26] and variations in community structure [27]. Many researchers have explored the effect of altitude on species composition, diversity, and forests formation structure during the previous two decades, with around half of these studies indicating an inverse association between species richness and altitude. Rahbek [30], on the other hand, did a quantitative investigation of altitudinal gradients of species richness and discovered that among plants, hump-shaped patterns of species diversity with peaks at mid-elevation are the most typically recorded, followed by monotonically declining patterns. Kluge et al. [31] found a hump-shaped diversity pattern for seed plants in the Eastern Himalayas, even though endemic species richness peaks at higher elevations due to increasingly isolated habitats and smaller surface area in mountainous ecosystems, which promotes speciation [32]. Although there has been a considerable increase in the number of phytosociological studies in these altitudinal regions considered hotspots of biodiversity [7, 19, 22, 23, 33], there is still limited knowledge of how and which environmental variables drive the distribution and composition of plant species along altitudinal gradients in specific hotspots regions, such as the Northwestern Himalaya.

By assessing the patterns of composition and distribution of plant species in these biodiversity hotspots influenced by environmental gradients, we greatly advance our understanding of the local plant community and how environmental factors are affecting these communities, which is a proxy for assessing how impacts of climate change can affect plant communities

located in mountainous regions [34]. In this context, we evaluated plant species composition and how and which environmental variables drive the plant species distribution of moist temperate zone of the Northwestern Himalaya, Pakistan. Briefly, we assessed (i) the potential plant communities present in the moist temperate zone; (ii) which are the environmental variables that most determine plant community structure in the moist temperate zone; (iii) whether there is variation in plant species composition between plant communities and which are the species that most contributed for species composition dissimilarity; (iv) whether there is variation of diversity indices among communities; and finally (v) which is the beta diversity process that most influence plant community structure in the moist temperate zone.

## Materials and methods

### Study area

The present study was carried out in the moist temperate zone of Manoor valley (District Mansehra, Khyber Pakhtunkhwa), which is a mountainous valley (34.68165 N to 34.83869 N latitude, and 73.57520 E to 73.73182 E longitude Fig 1) in the Northwestern Himalayan belt of Pakistan [35–39] along an altitudinal range of 1932–3168 m.a.s.l. Monsoon winds are the main source of precipitation as well as the primary force of controlling erosion, topography, climate and vegetation of the Northwestern Himalaya [1].

### Ethics approval and consent to participate

This study was approved by the Board of Study (BoS), Committee, Department of Botany and Advanced Study Research Board (ASRB) of Hazara University, Mansehra 21300, KP, Pakistan.

### Vegetation sampling and plant identification

In different growing seasons (from March to October), we evaluated 30 sampling sites per year, during four consecutive years, from 2015 to 2018. The line transect method (50 meters) we used for quantitative samplings [40–45], but we never repeated the same transects over years. The surveyed study area was subdivided into 30 stands and three points randomly selected were sampled within each sampling site along 50 meters transect (total = 90 transects).

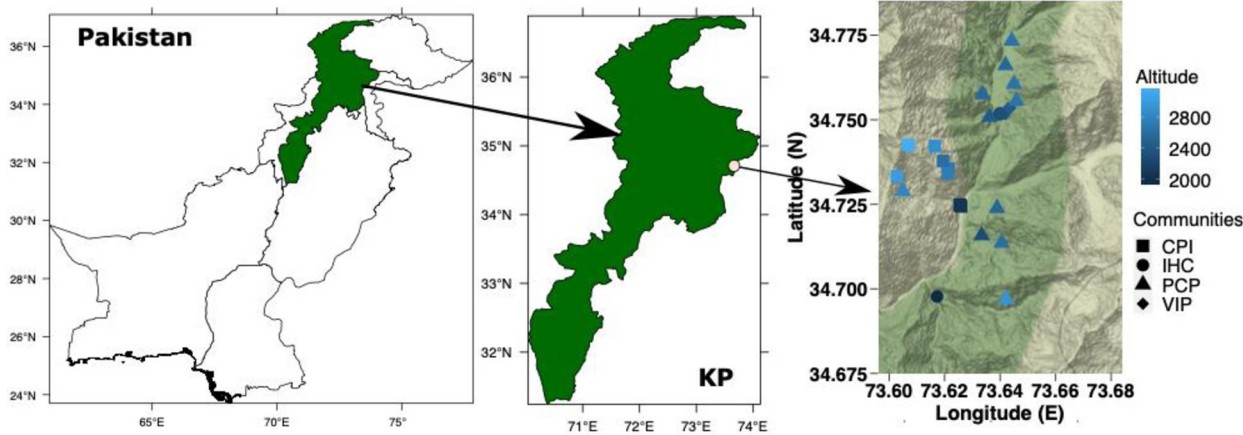

**Fig 1. Map of the study area showing Pakistan, Khyber Pakhtunkhwa (KP) province, and sampling sites for data collection.** Points in the right figure represent stands of the four communities identified in the moist temperate zone, Northwestern Himalaya, Pakistan. **CPI:** *Cedrus deodara-Pinus wallichiana-Isodon rugosus*, **IHC**: *Indigofera heterantha- Heracleum candicans-Cynodon dactylon*, **PCP:** *Pinus wallichiana-Cedrus deodara-Parrotiopsis jacquemontiana*, and **VIP:** *Viburnum grandiflorum-Indigofera heterantha-Pinus wallichiana*.

The distance between the sampling sites was kept at 200 meters, while the distance between the transects was kept at 100 meters. The individuals of plant species falling precisely on the line were recorded. The data from each sampling site was calculated using phytosociological characters (*i.e.*, density, frequency, cover and their relative values, as well as importance value) [46–48]. The IV was further used to rank each plant species and species with the highest IV were considered as the dominant species [46, 47]. Similarly, each plant community was named based on three dominant species [49–52]. The slope angle, aspect and exposure were recorded using clinometer; and altitude, longitude and latitude were measured by geographical positioning system (GPS). Plants species collection, labelling, pressing and other herbarium work methodology was adopted following Ijaz [53], Ijaz et al. [54], Amjad et al. [55] and Stefanaki et al. [56]. Identification was done with the aid of Flora of Pakistan [57–59] and submitted to the Herbarium of Hazara University, Mansehra (Pakistan).

## Environmental gradients

Soil samples weighting 200 grams were collected (15-30cm depth) randomly from each transect of sampled vegetation site [60, 61]. The replicated samples of each sampling site were thoroughly mixed to form a composite sample [62], which was placed in a sterile polythene bag and labelled accordingly. Raw materials such as rocks, and stones were sieved out and the samples were then shade dried. Each dried soil sample was processed for physicochemical tests [62, 63] to determine soil texture (*i.e.*, clay, sand, silt, loam), pH [64], electrical conductivity (EC) [65], organic matter (OM) [66], nitrogen (N), phosphorus (P), potassium (K) and calcium carbonate ($CaCO_3$) levels [60, 61, 67]. Other climatic variables such as barometric pressure, dew point, humidity, heat index, temperature, wet bulb (relative humidity and ambient air temperature) and wind speed were also determined using a small remote weather station (Kestrel weather tracker 4000) to record the data at each transect and then average values were calculated at sampling site level [19].

## Statistical analyses

The recorded data of vegetation, edaphic, and other environmental variables of sampling sites were compiled in order to determine relationship among them [68, 69]. Matrixes of IV data of all the recorded plant species (244 species) from 30 sampling sites were used in the analyses. Analyses were conducted on PC-ORD [70] and R software 4.0.0 [71]. Packages used in R software are mentioned in each analysis. A georeferenced map was prepared to show the distribution pattern of distinct plant communities (Fig 1).

**Sampling effort and cluster analyses.** Species area curves (SAC) were used to check the efficiency of the sampling effort, where plant abundance data with Sørensen distance values were used to create species area curves [72]. For classification of the recorded plant species (244) and 30 sampling sites into different plant communities, we used three different cluster analyses: Two-way Indicator Species Analysis (TWINSPAN), and Cluster Analysis (CA). We identified and classified plant species and stands (sampling sites) into major plant communities [73], as well as assess the effects of various factors (such as environmental variables) on such communities by processing clustering method using TWINSPAN [19, 74].

**Plant communities and associated environmental variables.** Both species and stands (sampling sites) were constrained in relation with the environmental gradients [75, 76], which were divided into geographic, slope aspect, edaphic and climatic gradients. We used non-metric multidimensional scaling (NMDS) and Principal Component Analysis (PCA) ordination biplot to determine the relationship between vegetation communities and environmental gradients using the "vegan" package. In NMDS and PCA, arrows represent the environmental

gradients, in which the length shows the strength of the gradient, and the direction represent the degree of correlation. The direction of gradients on the same axis reveals a positive correlation.

In addition, we performed canonical correspondence analysis (CCA) and variation partitioning tests (partial CCA) [77] to observe how explanatory variables (climatic, edaphic, geographic, and slope) drive the plant species distribution. First we built the best model with the lowest number of variables (those that most explain variance), through the *step* function with permutation using the "stats" package [71]. Next, we also evaluated multicollinearity between variables of the final model using Variance Inflation Factor (VIF), and we removed any variable with VIF>10, one at a time. Finally, with the final model, we carried out CCA and partial CCA to check how much each group of variables (geographic, edaphic, climatic, slope) explain in our model [77]. For these analysis, we used the "vegan" package [78].

**Variation of plant species composition among plant communities.** To compare whether there is difference in species composition between plant communities, we also used NMDS followed by a Permutational Multivariate Analysis of Variance (PERMANOVA) with Euclidean distance and 999 permutations. After PERMANOVA, we conducted pairwise comparisons between communities with corrections for multiple testing also using Euclidean distance and 999 permutations. We used false discovery rate (FDR) as p-value adjustment method. PERMANOVA and pairwise comparisons were conducted with "RVAideMemoire" package [79]. To observe the contribution of each plant species to overall dissimilarities, we used a similarity percentage analysis based on the decomposition of Bray-Curtis dissimilarity using the package "vegan" [78].

**Variation of diversity indices among plant communities.** To compare the diversity indices evaluated (species richness, Shannon, Simpson, and Pielou) among the four communities, we also conducted a GLM with Gaussian error distribution, except for species richness, in which we used Poisson distribution followed by Likelihood Ratio test. Pairwise comparisons were conducted with estimated marginal means using the package 'emmeans' [80].

**Beta diversity.** To evaluate which is the beta diversity process that most influence plant community structure in the moist temperate zone, we decomposed the Sørensen dissimilarity index (βsor), a measure of overall species replacement into two additive components: the spatial turnover (Simpson pairwise dissimilarity, βsim) and nestedness-resultant components (nestedness-fraction of Sorensen pairwise dissimilarity, βsne) [81–83]. Dissimilarity analysis was conducted in the package "betapart" [84].

## Results

### Sampling effort and plant communities

In total, 244 plant species belonging to 194 genera and 74 families (Table 1) were recorded in the moist temperate forests of the Manoor valley, Northwestern Himalaya, Pakistan. The moist temperate forests ranged from 1932.3m to 3168m. SAC analysis showed that the maximum number of plant species appeared up to stand 26 and the species curve became parallel after it, as no new species were recorded further (Fig 2).

Based on the TWINSPAN analysis (high cluster heterogeneity value–Lambda = 0.4045), we identified four different major plant communities (Fig 3), which were composed of different indicator species (IHC: *Indigofera heterantha-Heracleum candicans-Cynodon dactylon;* VIP: *Viburnum grandiflorum-Indigofera heterantha-Pinus wallichiana*, CPI: *Cedrus deodara-Pinus wallichiana-Isodon rugosus*, and PCP: *Pinus wallichiana-Cedrus deodara- Parrotiopsis jacquemontiana*). The IHC community was primarily found in the lower mountainous ranges (1932.3–2338.4 m.a.s.l), where the dominating flora was a combination of shrub and herb

**Table 1. Species composition and IV according to each sampling site and community found along four years of collection in moist temperate forests of Manoor valley, Northwestern Himalaya, Pakistan.**

| Plant Species | Abbreviations | Family name | Plant Communities | | | |
|---|---|---|---|---|---|---|
| | | | IHC | VIP | CPI | PCP |
| *Acer caesium* Wall. ex Brandis | ***Ace cae*** | Sapindaceae | 0.00 | 0.00 | 0.26 | 0.56 |
| *Achyranthes aspera* L. | ***Ach asp*** | Amaranthaceae | 0.00 | 0.00 | 0.00 | 0.34 |
| *Achyranthes bidentata* Blume | ***Ach bid*** | Amaranthaceae | 0.00 | 0.00 | 0.00 | 0.32 |
| *Achillea millefolium* L. | ***Ach mil*** | Asteraceae | 0.87 | 0.50 | 0.00 | 0.00 |
| *Adiantum capillus-veneris* L. | ***Adi cap-ven*** | Adiantaceae | 0.00 | 0.00 | 0.67 | 1.75 |
| *Adiantum indicum* J. Ghatak | ***Adi ind*** | Adiantaceae | 0.00 | 0.00 | 0.96 | 1.84 |
| *Adiantum venustum* D. Don | ***Adi ven*** | Adiantaceae | 0.00 | 0.00 | 0.49 | 1.30 |
| *Aegopodium burttii* Nasir | ***Aeg bur*** | Apiaceae | 0.00 | 0.00 | 0.15 | 0.34 |
| *Ainsliaea aptera* DC. | ***Ain apt*** | Asteraceae | 0.00 | 0.00 | 0.00 | 0.26 |
| *Ajuga integrifolia* Buch.- Ham. | ***Aju int*** | Lamiaceae | 0.00 | 0.00 | 0.00 | 0.27 |
| *Alchemilla cashmeriana* Rothum. | ***Alc cas*** | Rosaceae | 0.43 | 0.00 | 0.51 | 0.12 |
| *Alcea rosea* L. | ***Alc ros*** | Malvaceae | 0.05 | 0.00 | 0.15 | 0.00 |
| *Alnus nitida* (Spach) Endl. | ***Aln nit*** | Betulaceae | 0.45 | 0.00 | 0.00 | 0.29 |
| *Amaranthus viridis* L. | ***Ama vir*** | Amaranthaceae | 0.00 | 0.00 | 0.00 | 0.24 |
| *Anagallis arvensis* L. | ***Ana arv*** | Primulaceae | 0.00 | 0.00 | 0.08 | 1.15 |
| *Anaphalis busua* (Buch.-Ham.) DC. | ***Ana bus*** | Asteraceae | 0.00 | 0.00 | 0.20 | 0.07 |
| *Androsace hazarica* R.R. Stewart ex Y. Nasir | ***And haz*** | Primulaceae | 0.00 | 0.00 | 0.30 | 0.39 |
| *Androsace rotundifolia* Hardw. | ***And rot*** | Primulaceae | 0.00 | 0.00 | 0.06 | 0.54 |
| *Arisaema flavum* (Forssk.) Schott | ***Ari fla*** | Araceae | 0.00 | 0.00 | 0.00 | 0.91 |
| *Arisaema jacquemontii* Blume | ***Ari jac*** | Araceae | 0.00 | 4.24 | 0.00 | 0.91 |
| *Artemisia absinthium* L. | ***Art abs*** | Asteraceae | 0.36 | 0.13 | 0.96 | 2.25 |
| *Asplenium adiantum-nigrum* L. | ***Asp adi-nig*** | Adiantaceae | 0.00 | 0.00 | 0.20 | 0.41 |
| *Asparagus fiicinus* Buch.-Ham. ex D. Don | ***Asp fii*** | Asparagaceae | 0.00 | 0.00 | 0.00 | 0.16 |
| *Avena sativa* L. | ***Ave sat*** | Poaceae | 0.04 | 0.09 | 0.00 | 0.00 |
| *Bauhinia variegata* L. | ***Bau var*** | Caesalpiniaceae | 0.05 | 0.00 | 0.00 | 0.00 |
| *Bergenia ciliata* (Haw.) Sternb. | ***Ber cil*** | Saxifragaceae | 0.00 | 0.00 | 0.09 | 0.12 |
| *Berberis lycium* Royle | ***Ber lyc*** | Berberidaceae | 0.00 | 1.42 | 1.55 | 0.49 |
| *Berberis parkeriana* C.K. Schneid. | ***Ber pac*** | Berberidaceae | 0.00 | 0.00 | 0.00 | 0.24 |
| *Bistorta amplexicaulis* (D. Don) Greene | ***Bis amp*** | Polygonaceae | 5.17 | 1.64 | 0.00 | 0.00 |
| *Brassica compestris* L. | ***Bra com*** | Brassicaceae | 0.52 | 0.00 | 0.00 | 0.00 |
| *Bromus diandrus* Roth. | ***Bro dia*** | Poaceae | 2.00 | 1.64 | 0.39 | 0.19 |
| *Bromus secalinus* L. | ***Bro sec*** | Poaceae | 1.83 | 1.49 | 0.67 | 0.59 |
| *Bromus tectorum* L. | ***Bro tec*** | Poaceae | 2.20 | 1.29 | 0.00 | 0.06 |
| *Bupleurum longicaule* Wall. ex DC. | ***Bup lon*** | Apiaceae | 0.00 | 0.00 | 0.10 | 0.32 |
| *Bupleurum nigrescens* E. Nasir | ***Bup nig*** | Apiaceae | 0.51 | 1.22 | 0.15 | 1.48 |
| *Caltha palustris* var. *alba* (Cambess) Hook.f. & Thomson | ***Cal pal*** | Ranunculaceae | 0.00 | 0.00 | 0.04 | 0.46 |
| *Calamintha umbrosa* (M. Bieb.) Hedge | ***Cal umb*** | Lamiaceae | 1.11 | 1.03 | 1.08 | 0.61 |
| *Campylotropis meeboldii* (Schindl.) Schindl. | ***Cam mee*** | Papilionaceae | 0.00 | 0.00 | 0.32 | 0.04 |
| *Cannabis sativa* L. | ***Can sat*** | Cannabaceae | 0.00 | 0.00 | 0.00 | 0.07 |
| *Capsella bursa-pastoris* (L.) Medik. | ***Cap bur-pas*** | Brassicaceae | 0.18 | 0.97 | 0.53 | 1.47 |
| *Castanea sativa* Mill. | ***Cas sat*** | Fagaceae | 0.00 | 0.00 | 0.12 | 0.14 |
| *Cedrus deodara* (Roxb. ex Lamb.) G. Don | ***Ced deo*** | Pinaceae | 0.00 | 5.16 | 22.50 | 16.07 |
| *Celosia argentea* L. | ***Cel arg*** | Amaranthaceae | 0.61 | 0.00 | 0.00 | 0.00 |
| *Chenopodium album* L. | ***Che alb*** | Chenopodiaceae | 0.75 | 0.41 | 0.00 | 0.24 |
| *Chrysanthemum indicum* L. | ***Chr ind*** | Asteraceae | 0.00 | 0.00 | 0.00 | 0.64 |

*(Continued)*

**Table 1.** (Continued)

| Plant Species | Abbreviations | Family name | Plant Communities | | | |
|---|---|---|---|---|---|---|
| | | | IHC | VIP | CPI | PCP |
| *Cichorium intybus* L. | *Cic int* | Asteraceae | 0.00 | 0.00 | 0.23 | 0.11 |
| *Circaea alpina* L. | *Cir alp* | Onagraceae | 0.00 | 0.00 | 0.00 | 0.40 |
| *Cirsium arvense* (L.) Scop. | *Cir arv* | Asteraceae | 0.79 | 0.79 | 0.24 | 1.55 |
| *Circaea cordata* Royle | *Cir cor* | Onagraceae | 0.00 | 0.00 | 0.21 | 0.05 |
| *Cirsium falconeri* (Hook.f.) Petr. | *Cir fal* | Asteraceae | 0.00 | 0.00 | 0.00 | 0.07 |
| *Clematis grata* Wall. | *Cle gra* | Ranunculaceae | 0.00 | 0.00 | 0.96 | 2.55 |
| *Clinopodium vulgare* L. | *Cli vul* | Lamiaceae | 1.54 | 1.10 | 1.67 | 3.68 |
| *Colebrookea oppositifolia* Sm. | *Col opp* | Lamiaceae | 0.00 | 0.00 | 0.00 | 0.06 |
| *Commelina benghalensis* L. | *Com ben* | Commelinaceae | 0.00 | 0.65 | 0.00 | 0.13 |
| *Convolvulus arvensis* L. | *Con arv* | Convolvulaceae | 0.00 | 0.00 | 0.31 | 1.14 |
| *Conyza japonica* (Thunb.) Less. ex Less. | *Con jap* | Asteraceae | 0.00 | 0.18 | 0.09 | 0.40 |
| *Corydalis carinata* Lidén and Z.Y.Su | *Cor car* | Papaveraceae | 0.10 | 0.00 | 0.16 | 0.07 |
| *Corylus colurna* L. | *Cor col* | Betulaceae | 0.00 | 0.00 | 0.15 | 0.26 |
| *Corydalis cornuta* Royle [Syn. *Corydalis stewartii* Fedde] | *Cor cor* | Papaveraceae | 0.07 | 0.00 | 0.46 | 0.51 |
| *Cornus macrophylla* Wall. | *Cor mac* | Cornaceae | 0.05 | 0.00 | 0.00 | 0.23 |
| *Cornus oblonga* Wall. | *Cor obl* | Cornaceae | 0.00 | 0.00 | 0.00 | 0.19 |
| *Corydalis virginea* Lidén and Z.Y.Su | *Cor vir* | Papaveraceae | 0.00 | 0.00 | 0.13 | 0.18 |
| *Cotoneaster acuminatus* Wall. ex Lindl. | *Cot acu* | Rosaceae | 0.06 | 0.00 | 0.00 | 0.00 |
| *Cuscuta reflexa* Roxb. | *Cus ref* | Cuscutaceae | 0.00 | 0.00 | 0.08 | 0.34 |
| *Cyanthillium cinereum* (L.)H.Rob. | *Cya cin* | Asteraceae | 0.00 | 0.00 | 0.08 | 0.10 |
| *Cynoglossum apenninum* L. | *Cyn ape* | Boraginaceae | 0.08 | 0.18 | 0.00 | 0.00 |
| *Cynodon dactylon* (L.) Pers. | *Cyn dac* | Poaceae | 4.59 | 2.28 | 4.13 | 5.28 |
| *Cynoglossum glochidiatum* Wall. ex Benth. | *Cyn glo* | Boraginaceae | 1.58 | 0.25 | 0.33 | 1.52 |
| *Cynoglossum microglochin* Benth. | *Cyn mic* | Boraginaceae | 0.14 | 0.52 | 0.00 | 0.00 |
| *Cyperus odoratus* L. | *Cyp odo* | Cyperaceae | 0.15 | 0.89 | 0.08 | 0.20 |
| *Cyperus rotundus* L. | *Cyp rot* | Cyperaceae | 0.59 | 1.09 | 0.91 | 0.53 |
| *Dactylis glomerata* L. | *Dac glo* | Poaceae | 0.41 | 1.40 | 0.47 | 0.40 |
| *Daphne papyracea* Wall. ex G. Don | *Dap pap* | Thymelaeaceae | 0.00 | 0.00 | 0.06 | 0.25 |
| *Desmodium elegans* DC. | *Des ele* | Papilionaceae | 1.39 | 0.00 | 0.31 | 0.12 |
| *Dicliptera bupleuroides* Nees | *Dic bup* | Acanthaceae | 0.00 | 0.00 | 0.10 | 0.93 |
| *Dioscorea deltoidea* Wall. ex Griseb. | *Dio del* | Dioscoreaceae | 0.13 | 0.00 | 0.24 | 0.03 |
| *Diospyros lotus* L. | *Dio lot* | Ebenaceae | 0.00 | 0.00 | 0.16 | 0.00 |
| *Dipsacus inermis* Wall. in Roxb. | *Dip ine* | Dipsacaceae | 0.22 | 0.67 | 0.09 | 0.13 |
| *Dryopteris wallichiana* (Spreng.) Hyl. | *Dry wal* | Dryopteridaceae | 0.00 | 2.76 | 0.52 | 1.89 |
| *Duchesnea indica* (Andx) Fake. | *Duc ind* | Rosaceae | 0.00 | 0.00 | 0.00 | 0.54 |
| *Dysphania ambrosioides* (L.) Mosyakin & Clemants | *Dys amb* | Chenopodiaceae | 0.00 | 0.40 | 0.00 | 0.37 |
| *Elaeagnus umbellata* Thunb. | *Ela umb* | Eleagnaceae | 0.00 | 0.21 | 0.31 | 0.03 |
| *Elsholtzia ciliata* (Thunb.) Hyl. | *Els cil* | Lamiaceae | 0.00 | 0.16 | 0.18 | 0.04 |
| *Epilobium hirsutum* L. | *Epi hir* | Onagraceae | 0.04 | 0.00 | 0.17 | 0.12 |
| *Epilobium latifolium* L. | *Epi lat* | Onagraceae | 0.00 | 0.28 | 0.19 | 0.07 |
| *Epimedium elatum* C.Morrenand Decne. | *Epi ela* | Berberidaceae | 0.03 | 0.18 | 0.14 | 0.07 |
| *Equisetum arvense* L. | *Equ arv* | Equisetaceae | 0.40 | 0.00 | 0.00 | 0.00 |
| *Erigeron canadensis* L. | *Eri can* | Asteraceae | 0.87 | 0.00 | 0.55 | 0.39 |
| *Erysimum melicentae* Dunn. | *Ery mel* | Brassicaceae | 0.15 | 0.12 | 0.00 | 0.00 |
| *Euphorbia helioscopia* L. | *Eup hel* | Euphorbiaceae | 0.11 | 0.00 | 0.00 | 0.05 |
| *Euphrasia himalayica* Wetts. | *Eup him* | Orobanchaceae | 2.70 | 0.58 | 0.00 | 0.00 |

*(Continued)*

**Table 1.** (Continued)

| Plant Species | Abbreviations | Family name | Plant Communities | | | |
|---|---|---|---|---|---|---|
| | | | IHC | VIP | CPI | PCP |
| *Euphorbia hirta* L. | *Eup hir* | Euphorbiaceae | 0.00 | 0.00 | 0.09 | 0.25 |
| *Euphorbia prostrata* Ait. | *Eup pro* | Euphorbiaceae | 0.00 | 0.00 | 0.00 | 0.07 |
| *Euphorbia serpens* Kunth | *Eup ser* | Euphorbiaceae | 0.00 | 0.00 | 0.08 | 0.30 |
| *Fagopyrum tataricum* (L.) Gaertn. | *Fag tat* | Polygonaceae | 0.00 | 0.21 | 0.00 | 0.06 |
| *Filipendula vestita* (Wall. ex G. Don.) Maxim. | *Fil ves* | Rosaceae | 0.58 | 2.22 | 1.26 | 0.32 |
| *Foeniculum vulgare* Mill. | *Foe vul* | Apiaceae | 0.70 | 1.00 | 0.00 | 0.68 |
| *Fragaria nubicola* (Hook. f.) Lindl. ex Lacaita | *Fra nub* | Rosaceae | 0.15 | 1.95 | 0.36 | 4.11 |
| *Fumaria indica* (Hausskn) Pugsley | *Fum ind* | Fumaricaceae | 0.00 | 0.00 | 0.00 | 0.13 |
| *Galium aparine* L. | *Gal apa* | Rubiaceae | 0.00 | 0.00 | 0.00 | 0.01 |
| *Galium asparagifolium* Boiss. & Heldr. | *Gal asp* | Rubiaceae | 0.00 | 0.00 | 0.04 | 0.02 |
| *Galium elagans* Wall. | *Gal ela* | Rubiaceae | 0.00 | 0.00 | 0.05 | 0.13 |
| *Galinsoga parviflora* Cav. | *Gal par* | Asteraceae | 0.00 | 0.00 | 0.04 | 0.02 |
| *Gentianodes clarkei* (Kusn.) Omer | *Gen cla* | Gentianaceae | 0.00 | 0.00 | 0.00 | 0.10 |
| *Gerbera gossypina* (Royle) Beauverd | *Ger gos* | Asteraceae | 0.00 | 0.00 | 0.00 | 0.21 |
| *Geranium nepalense* Sweet. | *Ger nep* | Geraniaceae | 1.83 | 0.28 | 1.04 | 2.16 |
| *Geranium wallichianum* D. Don ex Sweet | *Ger wal* | Geraniaceae | 2.94 | 1.04 | 0.66 | 3.18 |
| *Gymnosporia royleana* Wall. ex M.A.Lawson | *Gym roy* | Celastraceae | 0.00 | 0.46 | 0.00 | 0.00 |
| *Hackelia uncinata* (Benth.) C.E.C. Fisch. | *Hac unc* | Boraginaceae | 0.33 | 0.00 | 0.00 | 0.00 |
| *Hedera nepalensis* K. Koch | *Hed nep* | Araliaceae | 0.00 | 0.00 | 0.00 | 2.61 |
| *Helianthus annuus* L. | *Hel ann* | Asteraceae | 0.14 | 0.00 | 0.00 | 0.00 |
| *Heracleum candicans* Wall. ex DC. | *Her can* | Apiaceae | 5.89 | 1.79 | 0.00 | 0.00 |
| *Hyoscyamus niger* L. | *Hyo nig* | Solanaceae | 0.00 | 0.00 | 0.07 | 0.00 |
| *Hypericum perforatum* L. | *Hyp perf* | Clusiaceae | 0.00 | 0.00 | 0.06 | 0.22 |
| *Impatiens bicolor* Royle. | *Imp bic* | Balsaminaceae | 0.12 | 0.29 | 2.10 | 3.56 |
| *Impatiens brachycentra* Kar. & Kir. | *Imp bra* | Balsaminaceae | 3.98 | 0.16 | 0.00 | 0.00 |
| *Indigofera australis* Willd. | *Ind aus* | Papilionaceae | 1.15 | 1.27 | 0.00 | 0.00 |
| *Indigofera hebepetala* Baker | *Ind heb* | Papilionaceae | 0.78 | 1.83 | 0.00 | 0.00 |
| *Indigofera heterantha* Brandis | *Ind het* | Papilionaceae | 5.08 | 23.51 | 2.33 | 2.83 |
| *Inula cuspidata* (Wall. ex DC.) C.B. Clarke | *Inu cus* | Asteraceae | 0.00 | 0.00 | 0.00 | 0.24 |
| *Inula falconeri* Hook.f. | *Inu fal* | Asteraceae | 0.00 | 0.00 | 0.05 | 0.10 |
| *Ipomoea nil* (L.) Roth | *Ipo nil* | Convolvulaceae | 0.00 | 0.00 | 0.26 | 0.50 |
| *Isodon rugosus* (Wall. ex Benth.) Codd | *Iso rug* | Lamiaceae | 0.00 | 0.00 | 5.77 | 3.32 |
| *Juglans regia* L. | *Jug reg* | Juglandaceae | 4.41 | 0.56 | 0.00 | 0.00 |
| *Lactuca tatarica* (L.) C.A. Mey | *Lac tat* | Asteraceae | 0.35 | 0.00 | 0.48 | 0.41 |
| *Lamium album* L. | *Lam alb* | Lamiaceae | 0.00 | 0.00 | 0.10 | 0.06 |
| *Lamium amplexicaule* L. | *Lam amp* | Lamiaceae | 1.23 | 0.25 | 0.53 | 0.88 |
| *Lathyrus aphaca* L. | *Lat aph* | Papilionaceae | 4.17 | 0.70 | 1.65 | 1.61 |
| *Lathyrus odoratus* L. | *Lat odo* | Papilionaceae | 0.30 | 0.74 | 0.00 | 0.00 |
| *Lathyrus sativa* L. | *Lat sat* | Papilionaceae | 0.34 | 0.90 | 0.00 | 0.00 |
| *Launaea procumbens* (Roxb.) Ramayya and Rajagopal | *Lau pro* | Asteraceae | 0.00 | 0.00 | 0.91 | 0.40 |
| *Lavatera cachemiriana* Camb. in Jacq. | *Lav cac* | Malvaceae | 0.04 | 0.00 | 0.00 | 0.00 |
| *Leptodermis virgata* Edgew. ex Hook.F. | *Lep vir* | Rubiaceae | 1.36 | 0.63 | 1.24 | 1.67 |
| *Ligularia amplexicaulis* DC. | *Lig amp* | Asteraceae | 0.00 | 0.34 | 0.00 | 0.11 |
| *Lindelofia sp.* | *Lin sp* | Boraginaceae | 0.05 | 0.00 | 0.00 | 0.00 |
| *Lomatogonium spathulatum* (A. Kern.) Fernald | *Lom spa* | Gentianaceae | 0.00 | 0.00 | 0.00 | 0.22 |
| *Lonicera caerulea* L. | *Lon cae* | Caprifoliaceae | 0.05 | 0.28 | 0.07 | 0.17 |

(*Continued*)

**Table 1.** (Continued)

| Plant Species | Abbreviations | Family name | Plant Communities | | | |
|---|---|---|---|---|---|---|
| | | | IHC | VIP | CPI | PCP |
| *Lotus corniculatus* L. | *Lot cor* | Papilionaceae | 0.00 | 0.14 | 0.03 | 0.07 |
| *Luffa sp.* | *Luf sp* | Cucurbitaceae | 0.00 | 0.00 | 0.39 | 0.37 |
| *Lyonia ovalifolia* (Wall.) Drude | *Lyo ova* | Ericaceae | 0.00 | 0.00 | 0.00 | 0.22 |
| *Malus domestica* Borkh. | *Mal dom* | Rosaceae | 0.27 | 0.35 | 0.05 | 0.22 |
| *Medicago sativa* L. | *Med sat* | Papilionaceae | 0.86 | 0.27 | 1.04 | 1.95 |
| *Mentha piperita* L. | *Men pip* | Lamiaceae | 0.00 | 0.00 | 0.00 | 0.64 |
| *Mentha royleana* Wall. ex Benth. | *Men roy* | Lamiaceae | 0.00 | 0.00 | 0.00 | 0.63 |
| *Micromeria biflora* (Ham.) Bth. | *Mic bif* | Lamiaceae | 0.00 | 0.00 | 2.17 | 0.88 |
| *Minuartia kashmirica* (Edgew.) Mattf. | *Min kas* | Caryophyllaceae | 0.00 | 0.00 | 0.00 | 0.12 |
| *Nepeta graciliflora* Benth. | *Nep gra* | Lamiaceae | 1.90 | 0.87 | 0.00 | 0.00 |
| *Nepeta laevigata* (D. Don) Hand.- Mazz | *Nep lae* | Lamiaceae | 1.00 | 2.07 | 0.00 | 0.00 |
| *Oenothera rosea* L. Her ex Aiton | *Oen ros* | Onagraceae | 1.36 | 0.46 | 0.26 | 0.44 |
| *Olea ferruginea* Wall. ex Aitch. | *Ole fer* | Oleaceae | 0.22 | 0.00 | 0.00 | 0.00 |
| *Onopordum acanthium* L. | *Ono aca* | Asteraceae | 0.00 | 1.57 | 0.00 | 0.00 |
| *Onychium contiguum* C. Hope | *Ony con* | Pteridaceae | 0.00 | 0.00 | 0.00 | 0.37 |
| *Origanum majorana* L. | *Ori maj* | Lamiaceae | 0.00 | 0.00 | 0.07 | 0.42 |
| *Origanum vulgare* L. | *Ori vul* | Lamiaceae | 0.00 | 0.00 | 0.74 | 1.06 |
| *Oxalis corniculata* L. | *Oxa cor* | Oxalidaceae | 1.28 | 0.24 | 1.89 | 4.94 |
| *Oxyria digyna* (L.) Hill | *Oxy dig* | Polygonaceae | 0.00 | 0.00 | 0.36 | 1.30 |
| *Parrotiopsis jacquemontiana* (Decne.) Rehder | *Par jac* | Hamamelidaceae | 0.43 | 0.00 | 4.69 | 10.33 |
| *Paspalum dilatatun* Poir. | *Pas dil* | Poaceae | 0.40 | 0.00 | 0.09 | 0.00 |
| *Pedicularis punctata* Decne | *Ped pun* | Orobanchaceae | 2.09 | 1.10 | 0.00 | 0.00 |
| *Pennisetum orientale* Rich. | *Pen ori* | Poaceae | 2.78 | 2.83 | 0.69 | 0.31 |
| *Periploca aphylla* Decne. | *Per aph* | Asclepiadaceae | 0.04 | 0.15 | 0.13 | 0.11 |
| *Persicaria capitata* (Buch.-Ham. ex D.Don) H.Gross | *Per cap* | Polygonaceae | 0.00 | 0.81 | 0.31 | 1.73 |
| *Pilea umbrosa* Blume | *Pil umb* | Urticaceae | 0.00 | 0.00 | 0.27 | 0.21 |
| *Pimpinella stewartii* (Dunn) Nasir | *Pim ste* | Apiaceae | 2.11 | 2.72 | 0.00 | 0.40 |
| *Pinus wallichiana* A.B. Jacks | *Pin wal* | Pinaceae | 0.00 | 5.30 | 20.28 | 16.24 |
| *Piptatherum aequiglume* (Duthie ex Hook.f.) Roshev. | *Pip aeq* | Poaceae | 0.13 | 0.00 | 0.00 | 0.00 |
| *Plantago lanceolata* L. | *Pla lan* | Plantaginaceae | 0.85 | 2.76 | 0.07 | 0.36 |
| *Plantago major* L. | *Pla maj* | Plantaginaceae | 2.06 | 4.64 | 0.71 | 0.79 |
| *Pleurospermum stellatum* (D. Don) Benth. ex C.B. Clarke | *Ple ste* | Apiaceae | 0.00 | 0.00 | 0.21 | 0.00 |
| *Pleurospermum stylosum* C.B. Clarke | *Ple sty* | Apiaceae | 0.00 | 0.00 | 0.24 | 0.04 |
| *Poa alpina* L. | *Poa alp* | Poaceae | 0.00 | 0.00 | 0.30 | 0.00 |
| *Poa annua* L. | *Poa ann* | Poaceae | 0.45 | 1.99 | 0.75 | 0.00 |
| *Poa infirma* Kunth | *Poa inf* | Poaceae | 3.08 | 3.03 | 1.11 | 0.03 |
| *Polygonum plebeium* R.Br. | *Pol ple* | Convallariaceae | 0.49 | 1.51 | 0.62 | 0.60 |
| *Polygonatum sp.* | *Pol sp.* | Convallariaceae | 0.00 | 0.00 | 0.00 | 0.13 |
| *Portulaca oleracea* L. | *Por ole* | Portulacaceae | 0.07 | 0.00 | 0.00 | 0.00 |
| *Potentilla anserina* L. | *Pot ans* | Rosaceae | 0.00 | 0.67 | 0.00 | 0.26 |
| *Potentilla nepalensis* Hook. | *Pot nep* | Rosaceae | 2.15 | 1.59 | 0.00 | 0.12 |
| *Prunus armeniaca* L. | *Pru arm* | Rosaceae | 0.07 | 0.00 | 0.00 | 0.00 |
| *Prunus cornuta* (Wall.ex Royle) Steud | *Pru cor* | Rosaceae | 0.25 | 0.00 | 0.00 | 0.00 |
| *Prunus domestica* L. | *Pru dom* | Rosaceae | 0.26 | 0.00 | 0.00 | 0.00 |
| *Prunella vulgaris* L. | *Pru vul* | Lamiaceae | 3.11 | 4.20 | 0.00 | 0.00 |
| *Pteridium aquilinum* (L.) Kuhn | *Pte aqu* | Pteridaceae | 0.29 | 0.00 | 0.00 | 0.00 |

*(Continued)*

**Table 1.** (Continued)

| Plant Species | Abbreviations | Family name | Plant Communities | | | |
|---|---|---|---|---|---|---|
| | | | IHC | VIP | CPI | PCP |
| *Pteracanthus urticifolius* (Wall. ex Kuntze) Bremek. | *Pte urt* | Verbenaceae | 0.00 | 0.00 | 0.07 | 0.05 |
| *Pteris vittata* L. | *Pte vit* | Pteridaceae | 1.34 | 0.00 | 0.51 | 0.49 |
| *Pyrus pashia* Buch.-Ham. ex D.Don | *Pyr pas* | Rosaceae | 0.69 | 0.00 | 0.00 | 0.00 |
| *Ranunculus laetus* Wall. ex Hook. f. and J.W. Thompson | *Ran lae* | Ranunculaceae | 0.00 | 0.00 | 0.00 | 0.07 |
| *Ranunculus muricatus* L. | *Ran mur* | Ranunculaceae | 1.61 | 0.94 | 0.00 | 0.23 |
| *Reinwardtia trigyna* Planch. | *Rei tri* | Linaceae | 0.00 | 0.00 | 0.04 | 0.04 |
| *Rhamnus purpurea* Edgew. | *Rha pur* | Rhamnaceae | 0.00 | 0.00 | 0.09 | 0.16 |
| *Rhynchosia pseudo-cajan* Cambess. | *Rhy pse* | Papilionaceae | 0.23 | 0.00 | 0.00 | 0.06 |
| *Rosa brunonii* Lindl. | *Ros bru* | Rosaceae | 0.00 | 0.00 | 0.15 | 0.07 |
| *Rosa webbiana* Wall. ex. Royle | *Ros web* | Rosaceae | 0.00 | 0.00 | 0.04 | 0.00 |
| *Rubus fruticosus* agg. | *Rub fru* | Rosaceae | 0.00 | 0.00 | 0.24 | 0.16 |
| *Rubus sanctus* Schreber | *Rub san* | Rosaceae | 0.00 | 0.00 | 0.00 | 0.12 |
| *Rumex dentatus* L. | *Rum den* | Polygonaceae | 0.47 | 0.00 | 0.00 | 0.00 |
| *Rumex nepalensis* Sprenge | *Rum nep* | Polygonaceae | 0.42 | 0.77 | 0.00 | 0.02 |
| *Rydingia limbata* (Benth.) Scheen & V.A. Albert [Syn. *Otostegia limbata* (Benth.) Boiss] | *Ryd lim* | Lamiaceae | 0.00 | 1.24 | 0.28 | 0.00 |
| *Saccharum spontaneum* L. | *Sac spo* | Poaceae | 0.00 | 0.00 | 0.00 | 0.28 |
| *Salvia lanata* Roxb. | *Sal lan* | Lamiaceae | 0.15 | 0.00 | 0.00 | 0.00 |
| *Salvia nubicola* Wall. ex Sweet | *Sal nub* | Lamiaceae | 0.12 | 0.00 | 0.00 | 0.00 |
| *Sanicula elata* Buch.-Ham. ex D.Don | *San ela* | Apiaceae | 0.00 | 0.00 | 0.10 | 0.25 |
| *Sambucus wightiana* Wall. ex Wight and Arn | *Sam wig* | Sambucaceae | 1.34 | 0.00 | 0.54 | 0.00 |
| *Sarcococca saligna* Müll.Arg. | *Sar sal* | Buxaceae | 0.00 | 0.00 | 0.00 | 1.54 |
| *Sauromatum venosum* (Dryand. ex Aiton) Kunth | *Sau ven* | Araceae | 0.00 | 0.00 | 0.00 | 0.11 |
| *Schismus arabicus* Nees. | *Sch ara* | Poaceae | 0.00 | 0.00 | 0.46 | 1.24 |
| *Senecio analogous* DC. | *Sen ana* | Asteraceae | 0.00 | 0.00 | 0.05 | 0.03 |
| *Senecio chrysanthemoides* DC. | *Sen chr* | Asteraceae | 0.00 | 0.00 | 0.15 | 0.57 |
| *Seseli libanotis* (L.) W.D.J. Koch . | *Ses lib* | Apiaceae | 0.00 | 0.00 | 0.04 | 0.31 |
| Sida cordata (Burm.f.) Borss.. | *Sid cor* | Malvaceae | 0.16 | 0.00 | 0.05 | 0.07 |
| *Silene conoidea* L. | *Sil con* | Caryophyllaceae | 0.37 | 0.00 | 0.00 | 0.05 |
| *Silene vulgaris* (Moench) Garcke | *Sil vul* | Caryophyllaceae | 0.37 | 0.23 | 0.00 | 0.17 |
| *Sisymbrium irio* L. | *Sis iri* | Brassicaceae | 0.03 | 0.32 | 0.05 | 0.00 |
| *Smilax glaucophylla* Koltzsch | *Smi glau* | Smilacaceae | 0.00 | 0.00 | 0.00 | 0.01 |
| *Solena amplexicaulis* (Lam.) Gandhi | *Sol amp* | Cucurbitaceae | 0.00 | 0.00 | 0.05 | 0.04 |
| *Sonchus asper* (L.) Hill | *Son asp* | Asteraceae | 0.00 | 0.00 | 0.00 | 0.00 |
| *Sorghum halepense* (L.) Pers. | *Sor hal* | Poaceae | 0.00 | 0.00 | 0.00 | 0.47 |
| *Sorbus tomentosa* Hedl. | *Sor tom* | Rosaceae | 0.00 | 0.00 | 0.00 | 0.06 |
| *Sorbaria tomentosa* (Lindl.) Rehder | *Sorb tom* | Rosaceae | 0.57 | 3.04 | 0.00 | 0.12 |
| *Spiraea affinis* R.Parker | *Spi aff* | Rosaceae | 0.00 | 0.00 | 0.00 | 0.12 |
| *Spiranthes sinensis* (Pers.) Ames | *Spi sin* | Orchidaceae | 0.00 | 0.00 | 0.00 | 0.02 |
| *Spiraea vaccinifolia* D. Don | *Spi vac* | Rosaceae | 0.00 | 0.00 | 0.09 | 0.02 |
| *Sporobolus diandrus* (Retz.) P.Beauv. | *Spo dia* | Poaceae | 2.48 | 0.00 | 0.75 | 0.26 |
| *Stellaria media* (L.) Vill. | *Ste med* | Caryophyllaceae | 0.11 | 0.00 | 0.00 | 0.18 |
| *Stellaria monosperma* Buch.-Ham. ex D. Don | *Ste mon* | Caryophyllaceae | 0.00 | 0.00 | 0.00 | 0.08 |
| *Swertia cordata* (Wall. ex G. Don) C.B. Clarke | *Swe cor* | Gentianaceae | 0.00 | 0.00 | 0.00 | 0.03 |
| *Tagetes minuta* L. | *Tag min* | Asteraceae | 0.00 | 0.00 | 0.70 | 2.22 |
| *Taraxacum officinale* aggr. F.H. Wigg. | *Tar off* | Asteraceae | 0.14 | 0.66 | 0.17 | 0.56 |
| *Thalictrum pedunculatum* Edgew. | *Tha ped* | Ranunculaceae | 0.00 | 0.00 | 0.13 | 0.05 |

*(Continued)*

**Table 1.** (Continued)

| Plant Species | Abbreviations | Family name | Plant Communities | | | |
|---|---|---|---|---|---|---|
| | | | IHC | VIP | CPI | PCP |
| *Torilis japonica* (Houtt.) DC. | *Tor jap* | Apiaceae | 0.00 | 0.00 | 0.05 | 0.08 |
| *Trachyspermum amii* (L.) Sprague | *Tra ami* | Apiaceae | 0.00 | 0.00 | 0.05 | 0.18 |
| *Trifolium repens* L. | *Tri rep* | Papilionaceae | 1.55 | 0.31 | 0.14 | 0.69 |
| *Urochloa panicoides* P. Beauv. | *Uro pan* | Poaceae | 0.61 | 0.43 | 0.30 | 0.00 |
| *Urtica dioica* L. | *Urt dio* | Urticaceae | 1.52 | 0.00 | 1.21 | 0.53 |
| *Valeriana jatamansi* Jones | *Val jat* | Caprifoliaceae | 0.00 | 1.10 | 0.00 | 0.00 |
| *Verbascum thapsus* L. | *Ver tha* | Scrophulariaceae | 1.34 | 0.00 | 0.00 | 0.73 |
| *Veronica anagallis* L. | *Ver ana* | Plantaginaceae | 0.00 | 2.12 | 0.23 | 0.24 |
| *Viburnum grandiflorum* Wall. ex DC. | *Vib gra* | Adoxaceae | 0.00 | 24.43 | 0.00 | 1.45 |
| *Vicia sativa* L. | *Vic sat* | Papilionaceae | 0.36 | 0.99 | 0.17 | 0.00 |
| *Vincetoxicum petrense* (Hemsl. & Lace) Rech. f. | *Vinc pet* | Asclepiadaceae | 0.00 | 0.00 | 0.09 | 0.13 |
| *Viola odorata* L. | *Vio odo* | Violaceae | 0.42 | 0.70 | 0.15 | 0.51 |
| *Viola serpens* Wall. Ex Ging | *Vio ser* | Violaceae | 0.19 | 0.92 | 0.38 | 0.11 |
| *Vitex negundo* L. | *Vit neg* | Vitaceae | 0.00 | 0.80 | 0.00 | 0.12 |
| *Wulfenia amherstiana* (Benth.) D.Y. Hong | *Wul amh* | Plantaginaceae | 0.00 | 0.00 | 0.04 | 0.34 |

**IHC**: *Indigofera heterantha- Heracleum candicans-Cynodon dactylon*, **VIP**: *Viburnum grandiflorum-Indigofera heterantha-Pinus wallichiana*, **CPI**: *Cedrus deodara-Pinus wallichiana-Isodon rugosus* and **PCP**: *Pinus wallichiana-Cedrus deodara- Parrotiopsis jacquemontiana*.

species owing to the existence of a substantial herbaceous layer of *Cynodon dactylon*, which carpeted the landscape alongside *Indigofera heterantha* patches (Table 1). The VIP community was recognized mainly in the foothills and adjacent plains (2390.5–2437.8 m.a.s.l), where the predominant vegetation was shrubland with abundant patches of *Viburnum grandiflorum* and

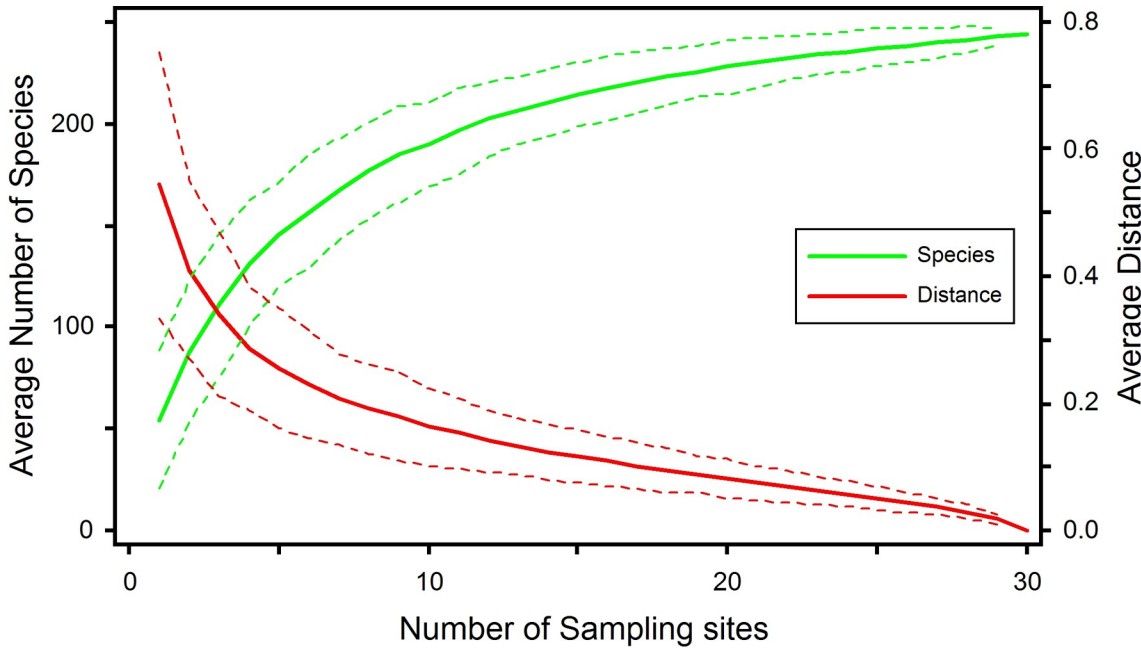

**Fig 2. The Species-Area Curve (SAC) of 244 plant species distributed among 30 sampling sites.** The SAC was used to check the adequacy level of the sampling effort, where plant abundance data with Sørensen distance values were used to create the SAC.

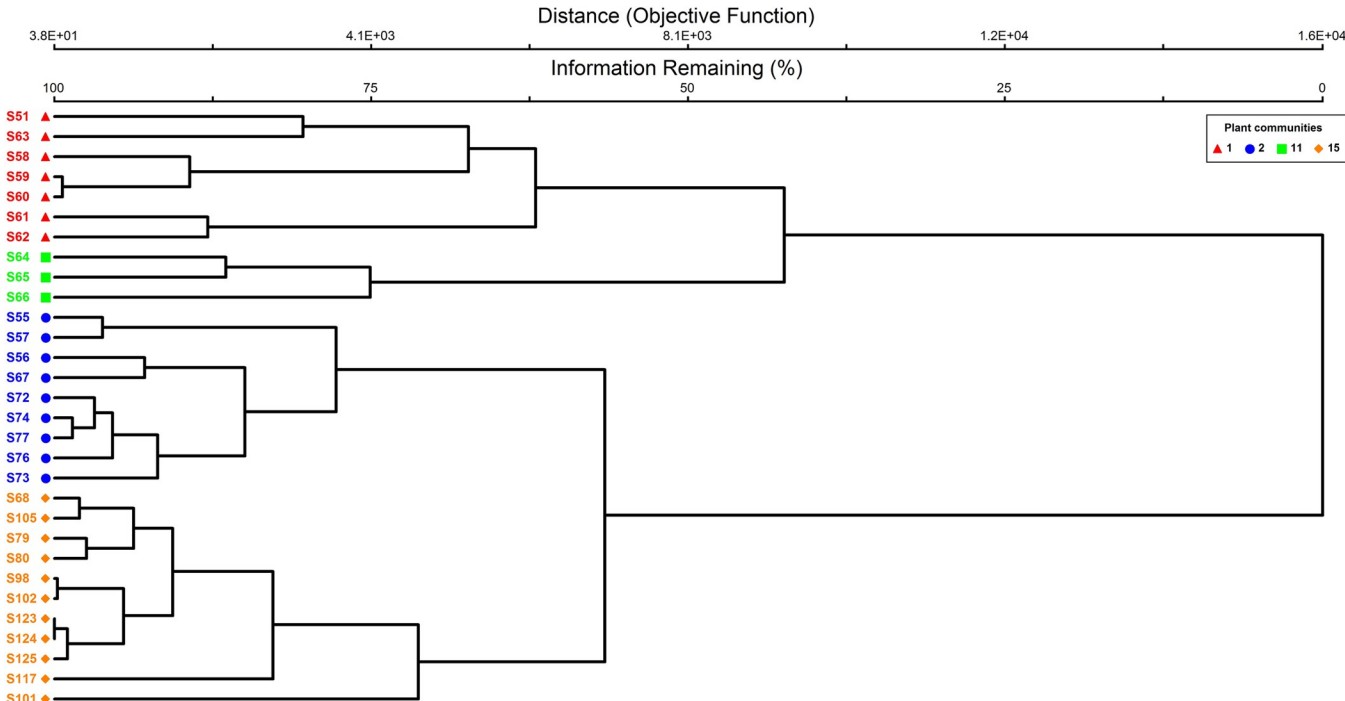

**Fig 3. Clustering analysis indicates the classification of 30 stands comprised of 244 plant species into four different plant communities.** IHC (red triangle): *Indigofera heterantha-Heracleum candicans-Cynodon dactylon*, VIP (blue circle): *Viburnum grandiflorum-Indigofera heterantha-Pinus wallichiana*, CPI (green square): *Cedrus deodara-Pinus wallichiana-Isodon rugosus* and PCP (yellow diamond): *Pinus wallichiana-Cedrus deodara- Parrotiopsis jacquemontiana*. The plant communities are represented by the symbols in the illustration. Letters associated with numbers at the end of each branch of the dendrogram represent the stands evaluated.

*Indigofera heterantha*, accompanied by the co-dominant *Pinus wallichiana* (tree species). Nonetheless, the other two plant communities, i.e., PCP and CPI, were significantly dominated by the tree species layer at the middle (2292–2947 m.a.s.l) and higher (2048.2–3168 m.a.s.l) altitudinal ranges alongside shrubby associates (Table 1).

## Plant communities and associated environmental variables

NMDS and PCA were used to show the relationship between the plant communities of moist temperate forests and environmental variables (Fig 4A–4D) and PCA (Fig 4E). The ecological and environmental variables like geographic, slope, edaphic, and climatic variables were used to correlate communities (Table 2). The most representative environmental variables that drive the community structure and diversity were altitude, slope angle and aspects (SE, NE, ES, WN), potassium (K), pH, organic matter, loam, silt, sand, clay, temperature, heat index, wind speed and barometric pressure. Environmental variables classify 30 sampling sites into four major plant communities, as shown by the cluster analysis (Fig 3). In constrained PCA ordination, the PC1 axis accounted for the most explanatory variance (20%), while the PC2 axis accounted for the least (14.2%). The profound influence of the environmental variables was revealed by classifying the moist temperate forests vegetation into four communities (Fig 4E), as also shown by CA, TWCA and NMDS.

The PCP community showed positive and significant correlation with northern aspect, silty loamy soil texture, humidity and altitude (Fig 4B and 4C). In contrast, CPI community showed positively significance with southern slope, wind speed, dew point and wet bulb. IHC community showed positive correlation with silty soil texture, electric conductivity and pH. And

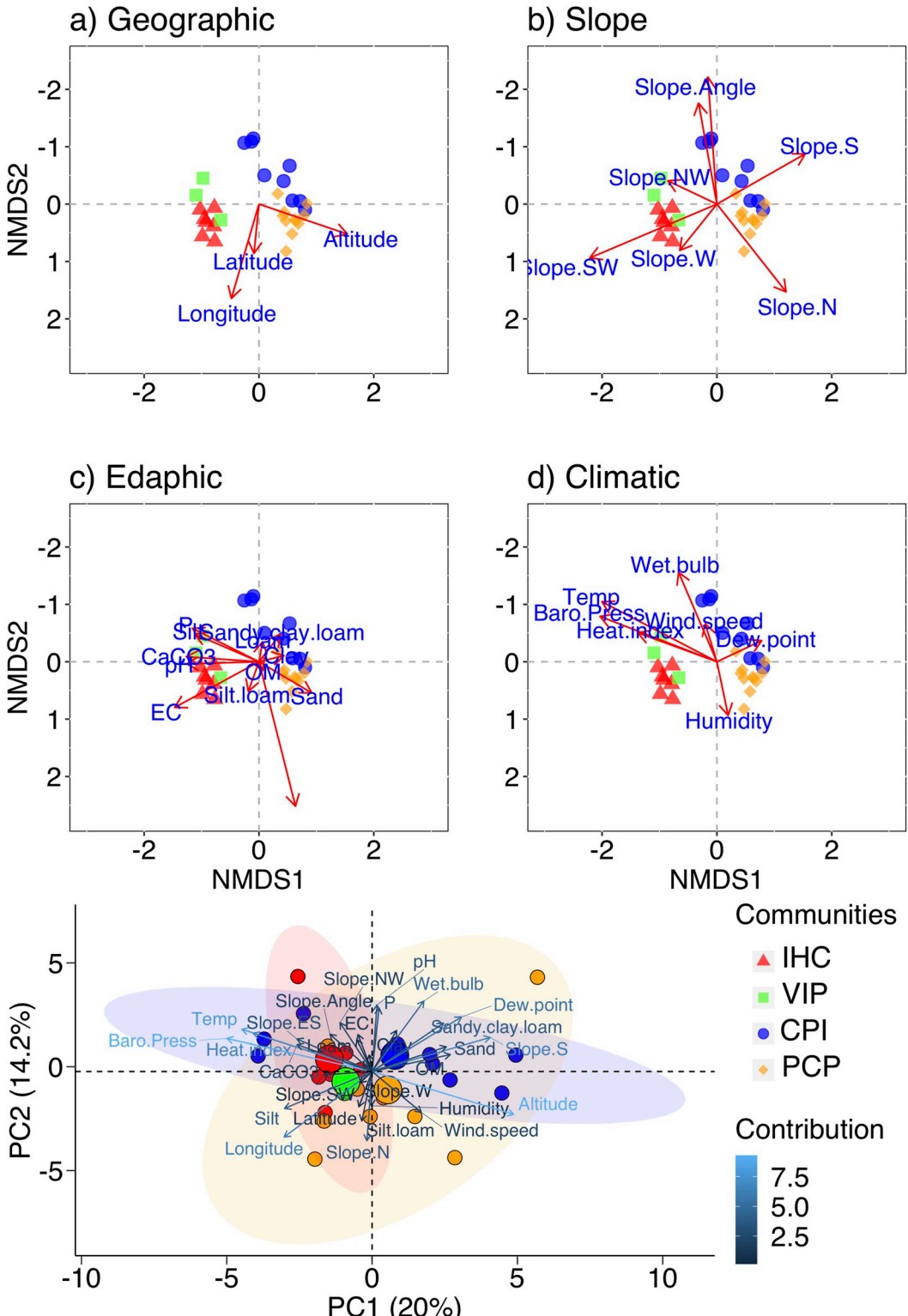

**Fig 4. Non-Multidimensional Scaling (NMDS) between plant communities in moist temperate forests and environmental gradients.** a) geographic, b) slope, c) edaphic and d) climatic. e). Principle Component Analysis (PCA) illustrating the relationship between various measured environmental variables and communities indicated by coloured circles. Large coloured circles show the centroid of each community. NMDS-PCA: Species contribution analysis for community ordination in NMDS is depicted in Table 2. **IHC:** *Indigofera heterantha- Heracleum candicans-Cynodon*

*dactylon*, **VIP:** *Viburnum grandiflorum-Indigofera heterantha-Pinus wallichiana*, **CPI:** *Cedrus deodara-Pinus wallichiana-Isodon rugosus* and **PCP:** *Pinus wallichiana-Cedrus deodara- Parrotiopsis jacquemontiana*.

finally, VPI community revealed positively significant correlation with north-western slope, $CaCO_3$ and pH. Thus, all the four communities were found separately in clumps with clear differences based on the environmental variables (Fig 4).

The CCA and variation partitioning tests showed that the total inertia results of CCA was 3.023, where our final variables (altitude, temperature, humidity, wind speed, slope angle, slope N, slope NW, slope SW, pH, EC, OM, $CaCO_3$, K, P, sand, and loam) together explained 66.5% of variation (sum of canonical eigenvalues was 2.011). The first two canonical axes explained 37.1% of variation. CCA model was significant ($\chi^2$ = 2.010; pseudo-F value = 1.613; p<0.001; df = 16; permutations = 999). For the 16 explanatory variables, we tested simple term

**Table 2. Mean (SD) of environmental variables and plant species richness per community found along four years of collection in moist temperate forests of Manoor valley, Northwestern Himalaya.**

| Communities | IHC | VIP | CPI | PCP |
|---|---|---|---|---|
| Species Richness | 51(8) | 53(10) | 40(12) | 68(13) |
| Altitude | 2251.7(132.7) | 2413(19.4) | 2588.8(408.8) | 2609(167.6) |
| Latitude | 34.7(0) | 34.8(0) | 34.7(0) | 34.7(0) |
| Longitude | 73.6(0) | 73.6(0) | 73.6(0) | 73.6(0) |
| Temp | 23.4(2) | 20.7(0.5) | 20.8(3.2) | 21(3) |
| Humidity | 56.8(6) | 54.6(3.7) | 54.7(3.6) | 56.7(3.7) |
| Heat index | 23.9(2.2) | 23.3(2.2) | 22.6(2.9) | 22.8(3.1) |
| Wind speed | 1.6(0.3) | 1.7(0.2) | 1.7(0.5) | 1.6(0.5) |
| Dew point | 16(0.9) | 16.3(0.5) | 16.5(1.5) | 16.6(2) |
| Wet bulb | 18.2(1.3) | 17.3(0.2) | 18.2(1.5) | 17.3(2.1) |
| Baro Press | 770.2(12.8) | 754.6(1.8) | 750.4(31.2) | 752.9(18.3) |
| Slope Angle | 47.9(16.9) | 35(4.1) | 56.6(31.7) | 46.7(22.2) |
| Slope ES | 0(0) | 0(0) | 0.3(0.5) | 0(0) |
| Slope N | 0(0) | 0(0) | 0(0) | 0.7(0.4) |
| Slope NW | 0.1(0.3) | 0(0) | 0(0) | 0(0) |
| Slope S | 0(0) | 0(0) | 0.7(0.5) | 0.3(0.4) |
| Slope SW | 0.9(0.3) | 0.7(0.5) | 0(0) | 0(0) |
| Slope W | 0(0) | 0.3(0.5) | 0(0) | 0(0) |
| pH | 5.8(0.2) | 5.6(0.2) | 5.6(0.5) | 5.4(0.5) |
| EC | 2.4(1.1) | 2(0.6) | 1.7(0.8) | 1.7(0.9) |
| OM | 1.2(0.3) | 1.3(0.3) | 1.3(0.5) | 1(0.4) |
| $CaCO_3$ | 6.3(1.6) | 9.3(1.9) | 6.6(2.4) | 5.6(2.4) |
| K | 210.9(5.6) | 220.3(5) | 210.9(3.1) | 216(5.2) |
| P | 13.4(3.2) | 11.7(0.5) | 11.9(3.2) | 10.5(3.8) |
| Sand | 31.2(3.6) | 27.6(2.8) | 30.5(8.3) | 35.2(6.9) |
| Silt | 46.5(6.1) | 46.7(3.5) | 44.3(7.5) | 41.7(7.6) |
| Clay | 22.4(4.1) | 25.7(1) | 25.2(2.4) | 23.2(4) |
| Loam | 0.6(0.5) | 0.3(0.5) | 0.6(0.5) | 0.5(0.5) |
| Sandy clay loam | 0(0) | 0(0) | 0(0) | 0.1(0.3) |
| Silt loam | 0.4(0.5) | 0.7(0.5) | 0.4(0.5) | 0.5(0.5) |

**IHC**: *Indigofera heterantha- Heracleum candicans-Cynodon dactylon*, **VIP:** *Viburnum grandiflorum-Indigofera heterantha-Pinus wallichiana*, **CPI:** *Cedrus deodara-Pinus wallichiana-Isodon rugosus* and **PCP:** *Pinus wallichiana-Cedrus deodara- Parrotiopsis jacquemontiana*.

effects. Simple term effects showed that Altitude, Slope SW, Slope NW, Slope N, Slope Angle, K, and Humidity (decreasing order of importance) were significant (p<0.05; Table 3). The 16 explanatory variables were grouped into four classes: Climatic (Humidity, Temperature, Wind speed); Edaphic (pH, EC, OM, CaCO₃, K, P, Sand, Loam); Geographic (Altitude); and Slope (Slope Angle, Slope N, Slope NW, Slope SW), and then, we performed variation partitioning tests (partial CCA) for all 15 possible classes (Table 4). Class [b] was the most explanatory variable (104.6%) followed by class [m] (7.2%) (Fig 5).

## Variation of plant species composition among plant communities and beta diversity

We found a significant variation in plant species composition among communities (Table 5; Fig 6), in which all communities showed a significant difference in species composition between each other (Table 6). Out of 244 species, six species greatly contributed to the variation in plant species composition between communities, namely *Viburnum grandiflorum*, *Indigofera heterantha*, *Heracleum candicans*, *Cedrus deodara*, *Pinus wallichiana*, and *Parrotiopsis jacquemontiana* (Table 6). Overall, the three species that most contributed for the variation in species composition between communities showed 13.7–29.7% of cumulative contribution (Table 6).

The total beta diversity (βsor) showed a value of 54.7% dissimilarity, of which spatial turnover (βsim) made up 40.5% and nestedness-resultant components (βsne) made up 14.2%. In βsim cluster, we observed 47.8% dissimilarity between PCP-CPI cluster and VIP-IHC cluster (Fig 7). PCP showed a dissimilarity of 9.4% with CPI, and VIP showed a dissimilarity of 21.5% with IHC (Fig 7). In βsne cluster, we found 24.5% dissimilarity between PCP and VIP-CPI-IHC cluster (Fig 7). VIP showed a dissimilarity of 11.3% with IHC-CPI, and IHC had 4.1%

**Table 3. The contribution and ranking of the studied variables in the variation partitioning tests (partial CCA model) to observe how explanatory variables (*i.e.*, climatic, edaphic, geographic, and slope) drive the plant species distribution.**

| Variables | Df | ChiSquare | F | p-value |
|---|---|---|---|---|
| Altitude | 1 | 0.251 | 3.229 | **0.001** |
| Slope.SW | 1 | 0.245 | 3.146 | **0.001** |
| Slope.NW | 1 | 0.178 | 2.297 | **0.001** |
| Slope.N | 1 | 0.214 | 2.748 | **0.002** |
| Slope.Angle | 1 | 0.163 | 2.097 | **0.009** |
| K | 1 | 0.140 | 1.798 | **0.018** |
| Humidity | 1 | 0.119 | 1.538 | **0.039** |
| Wind.speed | 1 | 0.114 | 1.472 | 0.069 |
| CaCO₃ | 1 | 0.098 | 1.270 | 0.151 |
| Temp | 1 | 0.081 | 1.049 | 0.376 |
| OM | 1 | 0.079 | 1.017 | 0.401 |
| EC | 1 | 0.076 | 0.976 | 0.474 |
| P | 1 | 0.070 | 0.907 | 0.578 |
| Sand | 1 | 0.069 | 0.895 | 0.590 |
| Loam | 1 | 0.056 | 0.724 | 0.864 |
| pH | 1 | 0.050 | 0.647 | 0.922 |
| P | 1 | 0.088 | 0.7295 | 0.820 |
| K | 1 | 0.075 | 0.6191 | 0.922 |

Significant variables are displayed in **bold**.

**Table 4. Results of variation partitioning tests (partial CCA model) of four environmental variable groups studied (*i.e.*, climatic, edaphic, geographic, and slope) that drives the plant species distribution.** For individual fraction letters code see Fig 5.

| Individual Fraction | Adjusted $R^2$ | Variation explained (%) | % of all | Df |
|---|---|---|---|---|
| [a] | 0.020 | 5.5 | 0.1 | 1 |
| [b] | 0.370 | 104.6 | 2.3 | 4 |
| [c] | 0.004 | 1.2 | 0.0 | 8 |
| [d] | 0.015 | 4.3 | 0.1 | 3 |
| [e] | 0.020 | 5.7 | 0.1 | 0 |
| [f] | -0.091 | -25.6 | -0.6 | 0 |
| [g] | -0.005 | -1.5 | 0.0 | 0 |
| [h] | -0.001 | -0.3 | 0.0 | 0 |
| [i] | -0.045 | -12.9 | -0.3 | 0 |
| [j] | -0.002 | -0.6 | 0.0 | 0 |
| [k] | 0.011 | 3.1 | 0.1 | 0 |
| [l] | 0.019 | 5.3 | 0.1 | 0 |
| [m] | 0.026 | 7.2 | 0.2 | 0 |
| [n] | 0.006 | 1.7 | 0.0 | 0 |
| [o] | 0.008 | 2.3 | 0.1 | 0 |
| Total explained | 0.354 | 100.0 | 2.2 | 18 |
| All variation | 15.835 | / | 100 | |

dissimilarity with CPI (Fig 7). Thus, plant community structure is twice more influenced by the spatial turnover of species (βsim) than by the species loss (nestedness-resultant, βsne).

## Variation of diversity indices among plant communities

We found a significant difference of four diversity indices, species richness (GLM $\chi^2$ = 73.113, df = 3, p<0.001; Fig 8A), Shannon (GLM $\chi^2$ = 35.797, df = 3, p<0.001; Fig 8B), Simpson

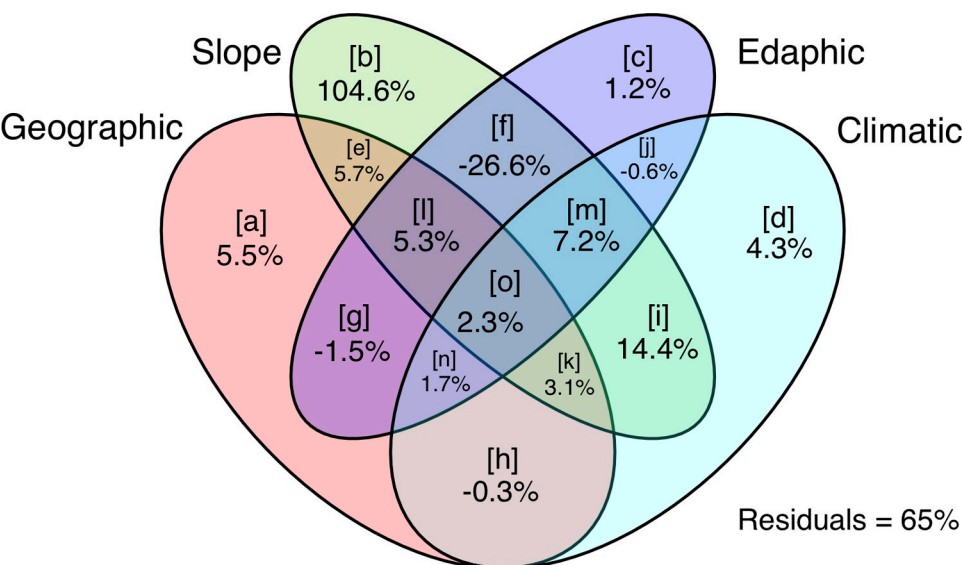

**Fig 5. The Venn diagram shows variation partitioning results (partial CCA model) and the contribution [77] of the four studied environmental variable groups (*i.e.*, climatic, edaphic, geographic, and slope) that drive the plant species distribution.** Each letter code indicates the individual fraction.

**Table 5. PERMANOVA results comparing species composition between the four communities found in Moist temperate forest.** This analysis was made with Euclidean distance and 999 permutations. Pairwise comparisons between communities are depicted in Table 6.

| | Df | Sums of Sqs | Mean Sqs | F | $R^2$ | Pr(>F) |
|---|---|---|---|---|---|---|
| Communities | 3 | 9961.1 | 3320.4 | 13.324 | 0.6059 | 0.001 |
| Residuals | 26 | 6479.1 | 249.2 | | 0.3941 | |
| Total | 29 | 16440.2 | | | 1 | |

(GLM $\chi^2$ = 46.465, df = 3, p<0.001; Fig 8C), and Pielou (GLM $\chi^2$ = 44.093, df = 3, p<0.001; Fig 8D), between the four communities. PCP showed the highest average number of species (68.1±4.2; mean±SE) followed by VIP (53.3±7.5) and IHC (51.1±3.5), and finally by CPI, with the lowest number of species (40.2±4.5) (Fig 8A). PCP showed a Shannon' value of 3.62±0.08 (mean±SE), followed by IHC (3.53±0.07), VIP (3.29±0.1), and CPI (2.9±0.1) respectively (Fig 8B). IHC showed the highest Simpson' value (0.959±0.01; mean±SE), followed by PCP (0.954 ±0.01), VIP (0.930±0.01), and CPI (0.898±0.01) respectively (Fig 8C). Finally, IHC showed the highest Pielou' value (0.901±0.01; mean±SE), followed by PCP (0.862±0.01), VIP (0.830±0.01), and CPI (0.797±0.01) respectively (Fig 8D).

## Discussion

Mountain ecosystems are characterized by dramatic changes in temperature and abiotic properties over short altitudinal and geographical distances, making them ideal natural laboratories for studying vegetation response to environmental changes [85]. In this study, we evaluated the plant species composition and distribution in a hotspot of biodiversity, the Northwestern Himalayan mountains, Pakistan, assessing how environmental gradients, source of habitat heterogeneity, influence plant community structure and diversity, which might be a proxy for assessing how climate change impacts on plant communities located in mountainous regions

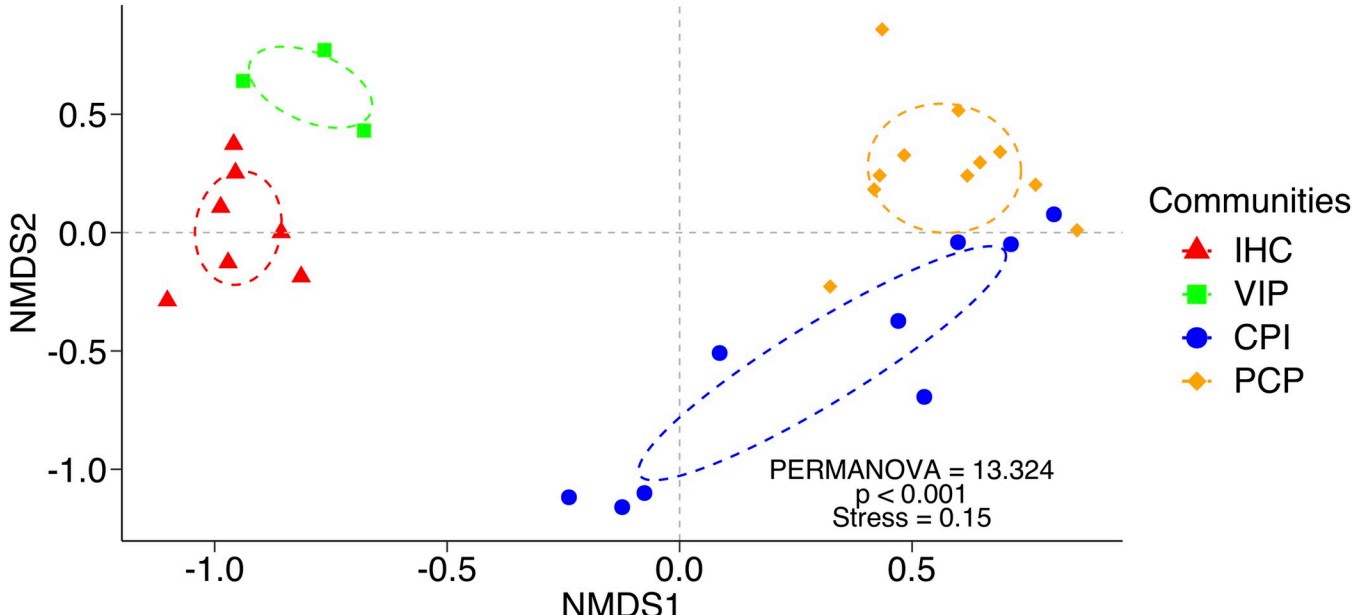

**Fig 6. NMDS with PERMANOVA analysis to compare species composition between communities of moist temperate forests. IHC**: *Indigofera heterantha-Heracleum candicans-Cynodon dactylon*, **VIP**: *Viburnum grandiflorum-Indigofera heterantha-Pinus wallichiana*, **CPI**: *Cedrus deodara-Pinus wallichiana-Isodon rugosus* and **PCP**: *Pinus wallichiana-Cedrus deodara- Parrotiopsis jacquemontiana*.

**Table 6. Pairwise comparisons with FDR p-value adjustment method of species composition and contrast results of the contribution of individual species to the overall Bray-Curtis dissimilarity of species composition between the four communities found in moist temperate forest.** We displayed only the three species that most contributed.

| Communities | P-value | Species | Av dis | SD | Ratio | Av Com1 | Av Com2 | Cum | Cum % | Cont % |
|---|---|---|---|---|---|---|---|---|---|---|
| **IHC-VIP** | 0.011 | Vib.gra | 0.1 | 0 | 5 | 0 | 24.4 | 0.1 | 12.2 | 12.2 |
| | | Ind.het | 0.1 | 0 | 1.9 | 8.8 | 23.5 | 0.2 | 20.1 | 7.9 |
| | | Her.can | 0 | 0 | 1.3 | 6.9 | 1.8 | 0.2 | 22.8 | 2.7 |
| **IHC-CPI** | 0.002 | Ced.deo | 0.1 | 0 | 3.8 | 0 | 22.5 | 0.1 | 10.9 | 10.9 |
| | | Pin.wal | 0.1 | 0 | 5.3 | 0 | 20.3 | 0.2 | 20.8 | 9.9 |
| | | Ind.het | 0 | 0 | 0.9 | 8.8 | 2.3 | 0.2 | 24.2 | 3.5 |
| **IHC-PCP** | 0.002 | Pin.wal | 0.1 | 0 | 2.7 | 0 | 16.2 | 0.1 | 6.9 | 6.9 |
| | | Ced.deo | 0.1 | 0 | 3 | 0 | 16.1 | 0.1 | 13.8 | 6.9 |
| | | Par.jac | 0 | 0 | 5.8 | 0.4 | 10.3 | 0.2 | 18 | 4.2 |
| **VIP-CPI** | 0.011 | Vib.gra | 0.1 | 0 | 5.1 | 24.4 | 0 | 0.1 | 11.6 | 11.6 |
| | | Ind.het | 0.1 | 0 | 5.3 | 23.5 | 2.3 | 0.2 | 21.5 | 9.9 |
| | | Ced.deo | 0.1 | 0 | 1.9 | 5.2 | 22.5 | 0.3 | 29.7 | 8.3 |
| **VIP-PCP** | 0.006 | Vib.gra | 0.1 | 0 | 3.3 | 24.4 | 1.4 | 0.1 | 9.5 | 9.5 |
| | | Ind.het | 0.1 | 0 | 6.6 | 23.5 | 2.8 | 0.2 | 17.9 | 8.4 |
| | | Pin.wal | 0 | 0 | 2.6 | 5.3 | 16.2 | 0.2 | 22.8 | 4.9 |
| **CPI-PCP** | 0.002 | Ced.deo | 0 | 0 | 1.2 | 22.5 | 16.1 | 0.1 | 5.4 | 5.4 |
| | | Par.jac | 0 | 0 | 1.8 | 4.7 | 10.3 | 0.1 | 9.6 | 4.2 |
| | | Pin.wal | 0 | 0 | 1.1 | 20.3 | 16.2 | 0.1 | 13.7 | 4 |

Av. dis.–Average dissimilarity; SD–Standard deviation; Av Com1 –Average Community 1; Av Com2 –Average community 2; Cum.–Cumulative; Cont.–Contribution.

Vib.gra: *Viburnum grandiflorum*, Ind.het: *Indigofera heterantha*, Her.can: *Heracleum candicans*, Ced.deo: *Cedrus deodara*, Pin.wal: *Pinus wallichiana*, Par.Jac: *Parrotiopsis jacquemontiana*.

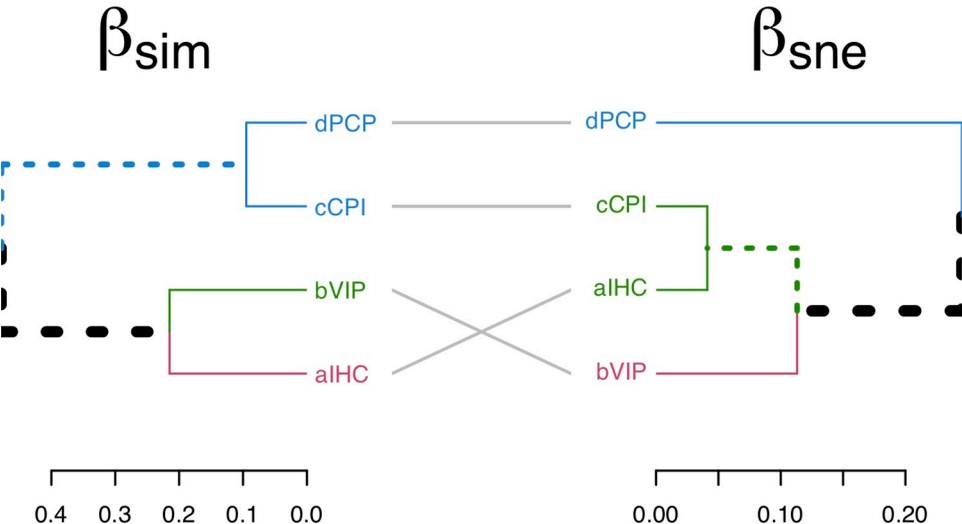

**Fig 7. Dissimilarity cluster based on spatial turnover (βsim) and nestedness-resultant components (βsne) of beta diversity components of species dissimilarity between four plant communities of moist temperate forests.** IHC: *Indigofera heterantha-Heracleum candicans-Cynodon dactylon*, VIP: *Viburnum grandiflorum-Indigofera heterantha-Pinus wallichiana*, CPI: *Cedrus deodara-Pinus wallichiana-Isodon rugosus*, and PCP: *Pinus wallichiana-Cedrus deodara-Parrotiopsis jacquemontiana*.

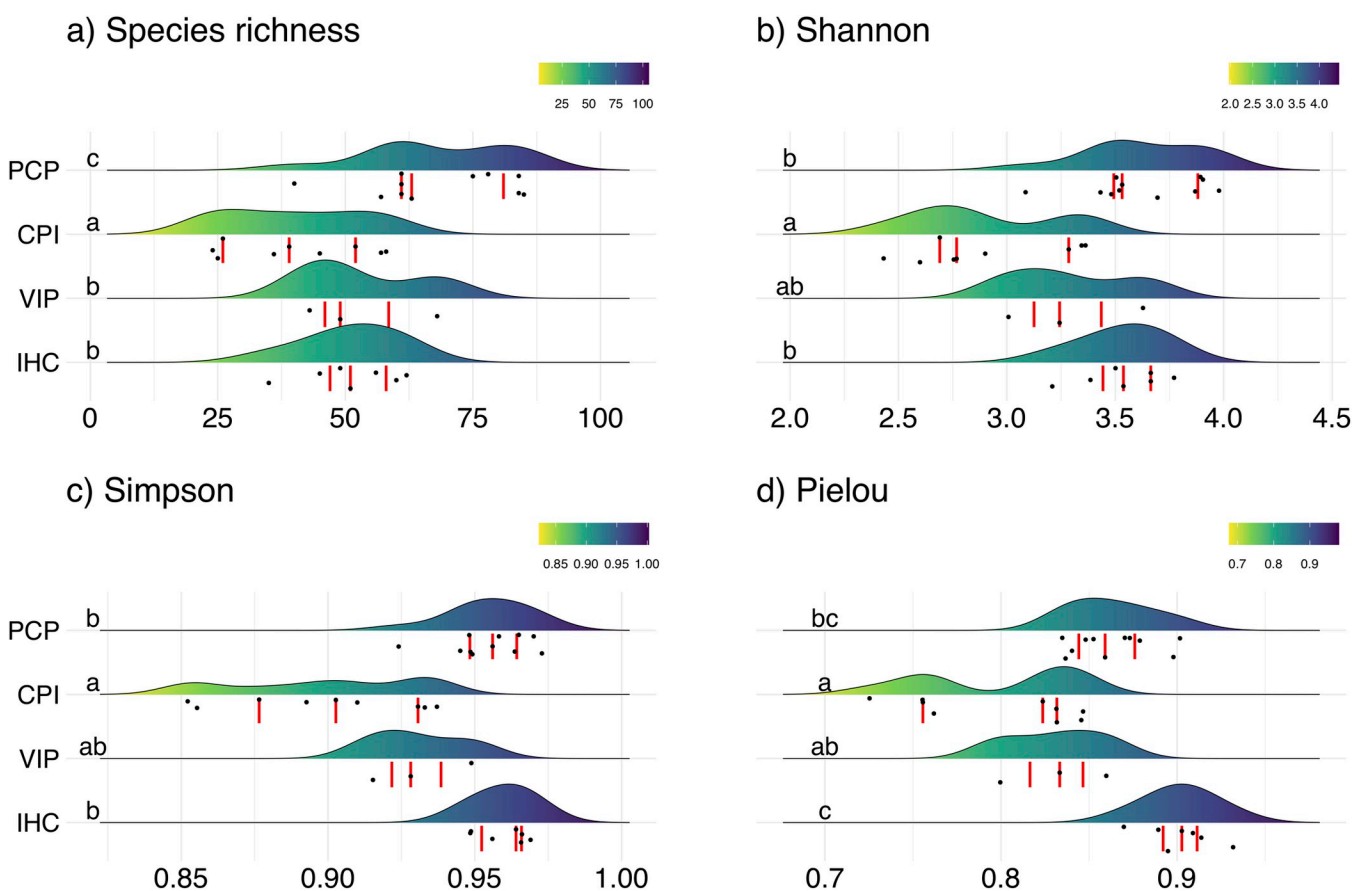

**Fig 8. Variation of diversity indices between the four plant communities of moist temperate forests in the Northwestern Himalaya, Pakistan.** Figures represent ridgeline plots with raw data (black dots below each density distribution) and the first, second and third quartiles (vertical red lines). Lowercase letters on the left differ from each other by an estimated marginal mean. The Y-axis is displayed in an ascendant altitudinal gradient. **IHC**: *Indigofera heterantha- Heracleum candicans-Cynodon dactylon*, **VIP**: *Viburnum grandiflorum-Indigofera heterantha-Pinus wallichiana*, **CPI**: *Cedrus deodara-Pinus wallichiana-Isodon rugosus*, and **PCP**: *Pinus wallichiana-Cedrus deodara- Parrotiopsis jacquemontiana*.

[34, 86, 87]. We found that (i) the moist temperate zone in this region can be divided in four different major plant communities; (ii) each plant community has a specific set of environmental drivers; (iii) there is a significant variation in plant species composition between communities, in which six species contributed most to the plant composition dissimilarity; (iv) there is a significant difference of the four diversity indices (species richness, Shannon, Simpson, Pielou) between communities; and finally (v) plant community structure is twice more influenced by the spatial turnover of species (βsim) than by the species loss (nestedness-resultant, βsne). Overall, we showed that altitudinal gradients offer an important range of different environmental variables, highlighting the existence of micro-climates that drive the structure and composition of plant species in each micro-region. In addition, each plant community along the altitudinal gradient has a set of environmental drivers, which lead to the presence of indicator species in each micro-region.

Mountain plant communities are thought to be sensitive to climate change and, thus, able to reveal its effects sooner than others [34, 88]. The four communities found showed a wide range of environmental drivers; however, altitude and temperature showed great prominence, probably making up the main environmental drivers in mountainous plant communities. Similar pattern was observed in the allied area (Nandiar catchment, Battagram) of Northwestern

Himalaya by stating altitude and temperature as the governing gradient [74]. Such variables, which can be strongly correlated [89], modify the diversity and structure of plant communities by creating local micro-climates [90], directly influencing plant community composition and diversity [19, 26, 91, 92].

Indeed, there is no order of importance of environmental variables, but studies are unanimous in showing that there is a consensus on the explanations for the variables' influences. For instance, in two recent studies we showed that the altitude-temperature relationship significantly influenced the physiological attributes of some plant species in the Northwestern Himalayan region [22, 23], which can be a proxy for understanding plant adaptation to climate change. Any change in soil parameters has a significant effect on the growth of plant communities [19]. The studies on mountain forests habitats around the world have also revealed the role of soil structure on species zonation [72, 93, 94]. Furthermore, both chemical and physical attributes of the soil are related to natural soil characteristics, with an impact on plant species composition and distribution of higher vascular plants [95–97]. For instance, some soil variables can have great influence on plant composition and distribution, such as pH. Some studies have shown that pH level on soil can influence nutrient availability, ultimately influencing nutrient uptake for growth [98–100]. However, the availability of some nutrients as a result of pH levels can be detrimental for some plants, since some nutrients are toxic to some plants [98, 101]. Considering that there is a great variability of pH levels and nutrient availability and concentration along altitudinal variables, it is expected a great variability of plant species composition which can be more or less related to specific soil parameters. Since plants are sensitive to small variations of soil characteristics such as pH, minerals, organic matter, among others, and these variables are constantly changing along altitudinal gradients directly and indirectly influencing the presence and availability of other organisms and resources, some plant species might have adapted to specific set of variables.

Variability in plant species diversity is an outcome of species interaction with particular set of environment variables either abiotic and biotic [102, 103], which can occur in both space and time [104, 105]. The concept of changing species composition and vegetation continuum along the ecological gradients emerged as an antithesis model for distinct units [106, 107]. In our study, the moist temperate forest of the studied Northwestern Himalayan region is comprised in an altitudinal gradient of approximately 1500 m. This gradient is subject to strong micro-climatic variation, which results in a set of micro-regions (better discussed above). Each micro-region has certain characteristics, which will influence the set of species that will inhabit these spaces [24–26]. In this sense, it is expected that the plant community structure is more influenced by the spatial turnover of species (βsim) than by the species loss (nestedness-resultant, βsne), *i.e.*, that there are different plant communities along the altitudinal gradient, as shown by our results. The differentiation of species diversity was mainly a consequence of environmental variables which is due to soil factors [108]. Therefore, in addition to the influence of edaphic factors in space on species composition and vegetation continuum, as shown in our study, results from similar studies have shown that the altitude is also important in driving vegetation structure and diversity in plant communities.

We found that environmental heterogeneity among plant communities have significant effects on beta diversity, particularly the spatial turnover. These results indicate that there is not a significant loss of the number of species between the plant community, but a variation in the species composition. This variation may be closely linked to the environmental effects in the area, which induces the appearance of species adapted to environmental variables [109]. The local community composition replacement implied the simultaneous loss and gain of species due to immigration–extinction dynamics and trait-based environmental filtering [110, 111]. This indicates the relationship among plant community types and among species based

on multiple factors. Although we did not find a large variation in βsne (loss of species between plant communities), it is important to note that temporal analysis might be important to consider a notable variation in this component of beta diversity; and βsne variations will be better observed in long-term analysis in future studies. In this sense, it is expected that the plant community structure is more influenced by the spatial turnover of species (βsim) than by the species loss (nestedness-resultant, βsne), *i.e.*, that there are different plant communities along the altitudinal gradient, as shown by our results. Similarly, results were report by Haq et al. [112] from forests of Kashmir Himalaya, India.

We observed a significant variation in plant species composition between communities, in which all communities showed a significant difference in species composition between each other. The measure of Bray-Curtis dissimilarity shows that species composition change that is influenced mainly by abundant species, in our study six species *(Viburnum grandiflorum, Indigofera heterantha, Heracleum candicans, Cedrus deodara, Pinus wallichiana*, and *Parrotiopsis jacquemontiana)* contributed most to the plant composition dissimilarity. These results suggest that the richness and turnover patterns we observed were driven primarily by rare species, which comprise most of the local species pools at these forest communities [113]. These findings are consistent with the idea that less abundant species are more sensitive to climate variability than longer lived and more abundant species [114]. The high level of turnover is common and is an important mechanism by which a large regional species pool buffers site level diversity from interannual variation in climate [115].

Current study provides the baseline and first insights of spatial distribution, vegetation pattern and species contribution in response to environmental gradients in a moist temperate forests, Northwestern Himalaya, Pakistan. Studies that evaluate the distribution and composition of the plant community are fundamental for a better understanding of the local plant community, the conservation status and protection of these communities, as well as providing support for mitigation measures. Especially in the case of Northwestern Himalaya, which represents a biodiversity hotspot, it is even more important that we conduct phytosociological studies in these areas to document and preserve the biodiversity there. In the face of current climate changes, these regions are being heavily impacted [28, 29], where the probability of species extinction may be higher than elsewhere, as these regions are rich in endemic species. Finally, we need to consider that phytosociological studies consider a general profile of the first trophic chains level, *i.e.*, to evaluate the composition, distribution and diversity of plants is to indirectly assess the first level of trophic chains.

## Author Contributions

**Conceptualization:** Inayat Ur Rahman, Aftab Afzal, Zafar Iqbal.

**Data curation:** Inayat Ur Rahman, Farhana Ijaz.

**Formal analysis:** Inayat Ur Rahman.

**Funding acquisition:** Inayat Ur Rahman.

**Investigation:** Inayat Ur Rahman.

**Methodology:** Inayat Ur Rahman, Farhana Ijaz.

**Project administration:** Robbie E. Hart, Aftab Afzal, Zafar Iqbal.

**Resources:** Inayat Ur Rahman, Elsayed Fathi Abd_Allah, Abdulaziz A. Alqarawi, Abeer Hashem.

**Software:** Inayat Ur Rahman, Eduardo S. Calixto.

**Supervision:** Robbie E. Hart, Aftab Afzal, Zafar Iqbal.

**Validation:** Robbie E. Hart, Aftab Afzal, Zafar Iqbal, Eduardo S. Calixto, Abdulaziz A. Alqarawi.

**Visualization:** Robbie E. Hart, Zafar Iqbal, Eduardo S. Calixto, Abeer Hashem.

**Writing – original draft:** Inayat Ur Rahman.

**Writing – review & editing:** Robbie E. Hart, Aftab Afzal, Zafar Iqbal, Eduardo S. Calixto, Elsayed Fathi Abd_Allah, Abdulaziz A. Alqarawi, Abeer Hashem, Al-Bandari Fahad Al-Arjani, Rukhsana Kausar, Shiekh Marifatul Haq.

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
