## [Decision Letter · Decision Letter 0]

24 May 2021

PONE-D-21-11989

Environmental variables drive plant species composition and distribution in the moist temperate forests of Northwestern Himalaya, Pakistan

PLOS ONE

Dear Dr. Rahman,

Thank you for submitting your manuscript to PLOS ONE. After careful consideration, we feel that it has merit but does not fully meet PLOS ONE’s publication criteria as it currently stands. Therefore, we invite you to submit a revised version of the manuscript that addresses the points raised during the review process.

We look forward to receiving your revised manuscript.

Kind regards,

Bhoj Kumar Acharya, PhD

Academic Editor

PLOS ONE

Additional Editor Comments:

The paper is well written with impressive datasets but lacks details in methodology, a robust discussion and conclusion. Please look into the comments given by all the three reviewers and address them well in the revised version of the MS. The reviewers have specifically commented on the methodology and discussion section which needs significant improvement. Additionally, please provide the list of plants along with details of their categories as supplementary materials. Although the introduction is well written but many important papers on elevational studies based on plants from the Himalaya (especially from Nepal and India) are not referred to. I suggest the authors to look into those literatures (minimum 10 such papers are are available) and try to give clear picture to what has already been done across the Himalaya in the subject.

Journal Requirements:

2) In your Methods section, please provide additional information regarding the permits you obtained for the work. Please ensure you have included the full name of the authority that approved the field site access and, if no permits were required, a brief statement explaining why.

3) Please amend your list of authors on the manuscript to ensure that each author is linked to an affiliation. Authors’ affiliations should reflect the institution where the work was done (if authors moved subsequently, you can also list the new affiliation stating “current affiliation:….” as necessary).

4) Your ethics statement should only appear in the Methods section of your manuscript. If your ethics statement is written in any section besides the Methods, please move it to the Methods section and delete it from any other section. Please ensure that your ethics statement is included in your manuscript, as the ethics statement entered into the online submission form will not be published alongside your manuscript.

Reviewers' comments:

Reviewer's Responses to Questions

**Comments to the Author**

1. Is the manuscript technically sound, and do the data support the conclusions?

Reviewer #1: Partly

Reviewer #2: Partly

Reviewer #3: No

2. Has the statistical analysis been performed appropriately and rigorously? 

Reviewer #1: Yes

Reviewer #2: Yes

Reviewer #3: Yes

3. Have the authors made all data underlying the findings in their manuscript fully available?

Reviewer #1: No

Reviewer #2: No

Reviewer #3: No

4. Is the manuscript presented in an intelligible fashion and written in standard English?

Reviewer #1: Yes

Reviewer #2: Yes

Reviewer #3: Yes

5. Review Comments to the Author

Reviewer #1: The basic data collection on vegetation, soil, etc. and analysis are completely lacking. Basic ecological expert may be consulted before finalizing the manuscript.

Overall, the environmental observation of the study is very less (4 years, 2015-2018) to draw proper conclusion on impact of plant diversity.

Relation to calculating the vegetation diversity along with environmental parameters, in location specific data needs clarification / rectification.

Further, the following points / comments need clarification / improvement.

Line no. 24-31: The abstract needs little improvement.

Line no. 32 & 100-104: The location can be provided with district / provinces / state, etc. in Pakistan.

Line no. 106: What are the growing seasons targeted? The time period of the study also be specified.

Line no. 116-118: What is the deepness of soil sampling? Whether the soil has been collected for each stands / transect, please clarify.

Line no. 119-121: The duration, intervals, etc. of data obtained from the handheld weather station and position / place of the weather station for the study are not clear.

Line no. 177: Details of 244 species recorded may be supplemented.

Line no. 182-186: The community may be specified as Trees, Shrubs, Grass, etc. and can be analysed properly. For example, the presence of Indigofera is shrub and Cynodon is grass and the landscape is shrubland or grassland. On the other hand, the presence of Pinus and Cedrus are clearly representing tree communities. Whether the landscape is having patches of vegetation of trees / shrub / grass composition or uniform vegetation dominated by these group, need clarification.

Line no. 190-194: Any specific observation on Cedrus, Pinus, etc. Because, we could not find the all vegetation in all aspects, especially the studied tree species.

Discussion: Need revision in view of answering the above questions.

Reviewer #2: The topic is timely, and such a kind of study, especially from the underrepresented areas, is essential and requires our attention. The authors have come up with clear questions and have analyzed them well. My main concern is with the methodology and discussion section; I see much basic information missing, especially concerning the data collection and study site; please look at each of the comments provided below. Discussion lacks proper explanation, key findings of different communities and predictor variables associated with them are interesting, but these points are not discussed well. Currently, the discussion is too general, but there is much potential to improve it. I also felt that your questions 2 and 4 explain the same points, so instead of keeping them as standalone objectives, it will be nice to combine them. In objective two results, all the four communities were found separately in clumps with apparent differences based on the environmental gradients, so I dint understand the need to again look into the variation of environmental variables among plant communities (iv). Lastly, instead of just reporting six species names, it will be good to have a table (primary or supplementary table), including all details of plant species in each community, their abundance, elevation range, etc.

Reviewer #3: MS number: PONE-D-21-11989

Title: Environmental variables drive plant species composition and distribution in the moist temperate forests of Northwestern Himalaya, Pakistan

General comments:

In this research paper, the authors studied influence of environmental variables in shaping plant species composition and their distribution in the moist temperate forests of Northwestern Himalaya, Pakistan. Covering 30 sampling sites, they measured 21 environmental variables for four consecutive years. Generated data is analysed using different multivariate analyses in order to identify potential plant communities and influence of environmental variable in species composition, distribution and diversity patterns. Overall the present research is well-designed and presents interesting results of multivariate analysis in order to understand how the existence of micro-climates drive the structure and composition of plant species in studied area.

Major issue:

1. In Conclusions (line 46-48) author stated that overall, we showed that altitudinal gradients offer an important range of different environmental variables, highlighting the existence of micro-climates that drive the structure and composition of plant species in each micro-region. However there is no mention of elevation range of study area across the manuscript.

2. Kindly add some detailed information on sampling method such as: if quadrat method is opted during the study what was the size of these used quadrats. Furthermore it will be good for readers to understand the study if information regarding the number of sampling quadrates in each transect will be mentioned in the methodology section.

3. A detailed table describing four communities (IHC; VIP; CPI; PCP) needs to be added highlighting species composition, environmental variables, elevation range, dominant families etc.

4. Page 9 line 177 author mentioned “A total of 244 plant species were recorded 177 in moist temperate forest of Manoor valley, Himalaya, Pakistan”. Kindly add details of 244 plant species belonging to ……………genus and …….families respectively.

Other comments:

1. Kindly follow uniformity across the manuscript text to avoid confusion, such as spelling variations (in line 122 analyses and 123 analyzes) and such omissions needs to be checked throughout the manuscript.

2. Author has used the term Northwestern Himalaya (line 23); Himalaya (line 31): Northwestern Himalaya (line 102); Himalayan mountains (line 260) etc. thus it is suggested to use term Northwestern Himalaya to indicate study site throughout the manuscript.

3. For wider acceptability of the manuscript it is suggested to discuss similar studies carried out in other parts of Himalaya in discussion section.

6. PLOS authors have the option to publish the peer review history of their article (what does this mean?). If published, this will include your full peer review and any attached files.

Reviewer #1: **Yes: **K. Chandra Sekar

Reviewer #2: No

Reviewer #3: **Yes: **Dr. Aseesh Pandey

---

## [Author Response · Author response to Decision Letter 0]

12 Aug 2021

Revision notes

Reviewer’s comments black font; author answers in blue font; responses are numbered (R#1, R#2, etc. for reference)

Reviewer(s)’ Comments to Author:

Reviewer #1

General comments:

In this research paper, the authors studied influence of environmental variables in shaping plant species composition and their distribution in the moist temperate forests of Northwestern Himalaya, Pakistan. Covering 30 sampling sites, they measured 21 environmental variables for four consecutive years. Generated data is analysed using different multivariate analyses in order to identify potential plant communities and influence of environmental variable in species composition, distribution and diversity patterns. Overall the present research is well-designed and presents interesting results of multivariate analysis in order to understand how the existence of micro-climates drive the structure and composition of plant species in studied area.

R#1 – We really appreciate the positive words.

Major issue:

1. In Conclusions (line 46-48) author stated that overall, we showed that altitudinal gradients offer an important range of different environmental variables, highlighting the existence of micro-climates that drive the structure and composition of plant species in each micro-region. However there is no mention of elevation range of study area across the manuscript. 

R#2 – We now added this information in the manuscript (L.14-15,93,237-246). The altitudinal range of our data sampling ranged from 1932 to 3168 m.a.s.l. 

2. Kindly add some detailed information on sampling method such as: if quadrat method is opted during the study what was the size of these used quadrats. Furthermore it will be good for readers to understand the study if information regarding the number of sampling quadrates in each transect will be mentioned in the methodology section. 

R#3 – Required information regarding the vegetation sampling has been added. “The surveyed stands (study area) were subdivided into 30 stands and three points randomly selected within each stand were sampled along 50 meters transect (total = 90 transects). The interval distance kept between the stands was 200 meters and 100 meters between transects.” (L. 97-110)

3. A detailed table describing four communities (IHC; VIP; CPI; PCP) needs to be added highlighting species composition, environmental variables, elevation range, dominant families etc.

R#4 – We now added two new supplementary tables (Table S1 and S2) describing the communities. One table describes environmental variables and the other table the species composition for each plant community. 

4. Page 9 line 177 author mentioned “A total of 244 plant species were recorded 177 in moist temperate forest of Manoor valley, Himalaya, Pakistan”. Kindly add details of 244 plant species belonging to ……………genus and …….families respectively. 

R#5 – As per suggestion, the required details has been added i.e., 194 genera and 74 families. (L. 220)

Other comments: 

1. Kindly follow uniformity across the manuscript text to avoid confusion, such as spelling variations (in line 122 analyses and 123 analyzes) and such omissions needs to be checked throughout the manuscript. 

R#6 – Done. Thank you. (L. 15, 139, 146, 219, 294)

2. Author has used the term Northwestern Himalaya (line 23); Himalaya (line 31): Northwestern Himalaya (line 102); Himalayan mountains (line 260) etc. thus it is suggested to use term Northwestern Himalaya to indicate study site throughout the manuscript.

R#7 – We changed to Northwestern Himalaya in the places that we are talking about the specific region of Himalayas. Other parts are related to the Himalayas in a general perspective and then we can change it. We appreciate your understanding. (L. 10, 12, 40, 71, 80, 92, 95 and onwards)

3. For wider acceptability of the manuscript it is suggested to discuss similar studies carried out in other parts of Himalaya in discussion section. 

R#8 – We added more citations about studies conducted in the Himalayan mountains. We have now discussed it in detail. We now also added more citations about plant community studies in the Himalayas. (L. 327-329, 367-369, 376-382, 398-432)

Reviewer #2

The topic is timely, and such a kind of study, especially from the underrepresented areas, is essential and requires our attention. The authors have come up with clear questions and have analyzed them well. 

R#9 – We really appreciate your positive words. 

My main concern is with the methodology and discussion section; I see much basic information missing, especially concerning the data collection and study site; please look at each of the comments provided below. Discussion lacks proper explanation, key findings of different communities and predictor variables associated with them are interesting, but these points are not discussed well. Currently, the discussion is too general, but there is much potential to improve it. I also felt that your questions 2 and 4 explain the same points, so instead of keeping them as standalone objectives, it will be nice to combine them. In objective two results, all the four communities were found separately in clumps with apparent differences based on the environmental gradients, so I dint understand the need to again look into the variation of environmental variables among plant communities (iv). Lastly, instead of just reporting six species names, it will be good to have a table (primary or supplementary table), including all details of plant species in each community, their abundance, elevation range, etc.

R#10 – We now improved our manuscript and cordially ask you to check our responses for each question raised. Overall, we added more information in methods (L. 93, 97-110, 112-114, 121-125, 136-138, 140-142, 144-145, 148-156) and discussion (L. 327-329, 367-369, 376-382, 398-432) sections. Following reviewer’s very important suggestion, two new supplementary tables with a summary of the environmental variables and the species composition based on the importance value were also added. 

L39-40: ii) each plant community has a specific set of environmental drivers & L41-42 iv) most of the environmental variables were significantly different between communities. I feel both of these sentences highlight the same points; my advice would be to combine both and write them as a single point.

R#11 – Thank you for your suggestion. We agreed and combined accordingly. (L. 22, 84, 339)

L90-91- and L94: I am highlighting this point again; there is an overlap in your questions 2 and 4

R#12 – Please see R#11 for the answer about this question. (L. 22, 84, 339)

L106 What were the growing seasons? Please mention the seasons name

R#13 – We now added the months that represent the growing seasons in the area, which were from March to October. (L. 97)

L106 Was the same stand sampled for four years, meaning four times? Please note that many important information on sampling design is missing; for example, how were the individuals counted, those falling precisely on the line? Furthermore, was the line horizontal or vertical? Was it stratified random sampling? Also, what was the distance between the two lines transect?

R#14 – Yes. We repeated the stands over years, but not the same transects inside each stand. We now added more information about it the text. The surveyed stands (study area) were subdivided into 30 stands and three points randomly selected within each stand were sampled along 50 meters transect (total = 90 transects). The interval distance kept between the stands was 200 meters and 100 meters between transects. The individuals of plant species falling precisely on the line were noted. (L. 98-110)

L116 There is no clarity on how soil samples, Ph values were collected. Was it collected at a single location per stand, and what depth, slope, well-exposed sunlit areas or not? 

Ph varies a lot, even at a short distance between north and south facing slopes, so it is better to provide all the sampling details.

R#15 – Basic details regarding the data collection of vegetation and soil sampling are now mentioned in Vegetation sampling (L. 98-110, 112-114), and Environmental gradients (L. 121-125, 136-138) respectively. A supplementary table S2 has also been included to show the variability of environmental gradients among the four plant communities.

L121 At what distance was the weather station located from those 30 stands? The environmental variable varies across the elevation, and authors have also mentioned the importance of microclimates, so I am curious to know how many weather stations were installed. 

The overall sampling design will become more apparent if the authors explain how the line transects and weather stations are distributed along the elevation gradient.

R#16 – We used a small remote weather station and recorded the data at transect level and then averaged it to stand level. Required details regarding vegetation sampling using line transect are incorporated into the Vegetation sampling section (M & M).

Results:

I feel that apart from the given figures, it will be good to have a table where the species names, elevation range, abundance in each community are mentioned. A comprehensive table with all these details may give a clearer picture of your findings. You may decide how much information you would want to share in the main manuscript and supplementary information.

R#17 – Thank you for your comment. We now added two supplementary tables about plant species composition and environmental variables values in each community. Since we already have many tables and figures in the manuscript, and these other tables will be so long, we are adding them as supplementary tables (S1 and S2).

Fig 3: Legend need to be rewritten, the author has mentioned four communities' names and four colors corresponding to each community, but the link between each color and community is missing. Also, what do the alphabet and various numbers indicate on the left side of the graph?

R#18 – We now updated our legend. Numbers and letter are the stands.

“Figure 3. Clustering method using two way indicator species analysis (TWINSPAN) indicating four different plant communities. IHC (red triangle): Indigofera heterantha-Heracleum candicans-Cynodon dactylon, VIP (blue circle): Viburnum grandiflorum-Indigofera heterantha-Pinus wallichiana, CPI (green square): Cedrus deodara-Pinus wallichiana-Isodon rugosus and PCP (yellow diamond): Pinus wallichiana-Cedrus deodara- Parrotiopsis jacquemontiana. Letters associated to numbers at the end of each branch of the dendrogram represent the stands evaluated. 

L209: What was the intention behind considering temperature and altitude both as the predictor variables? Because along the mountain gradient these variables are highly correlated.

R#19 – In this specific analysis, we did not consider any kind of multicollinearity. We plot all variables to see the overall relationship between them and communities. However, if you check the partial CCA and partitioning analyses, you will see that we removed the colinear variables. Therefore, in the first analysis (NMDS, PCA) we show the overall relationship of all variables with each community, while in the second analysis, we show which are the variables that most explain the patterns found in our study.

Discussion: In the result, section author has emphasized their findings of four distinct communities and the environmental variables associated with each one of them. However, in the discussion, the authors have failed to discuss their unique findings in detail. I feel that the discussion section is too general and poorly explained. There is a lot of scopes to improve the discussion considering the amount of analysis carried out. By highlighting just temperature and altitude (which is definitely important from the climate change aspect), you may lose other valuable details of your study.

R#20 – We totally agree. In discussion, we provided more details about how these other variables can influence plant community. We believe that discussing each variable and its relationship to the community will extend a lot the manuscript, turning it tiring for readers. In this context, we believe that the general patterns and discussion of the main variables should be the best approach. We appreciate your comment, and we cordially ask to check the lines mentioned, where we added information about the influence of other variables in the community structure. Some important edaphic variables like pH were considered and discussed accordingly. (L. 327-329, 367-369, 376-382, 398-432)

Reviewer #3

Comments on the manuscript entitled ‘Environmental variables drive plant species composition and distribution in the moist temperate forests of Northwestern Himalaya, Pakistan’. 

The basic data collection on vegetation, soil, etc. and analysis are completely lacking.

Overall, the environmental observation of the study is very less (4 years, 2015-2018) to draw proper conclusion on impact of plant diversity.

Relation to calculating the vegetation diversity along with environmental parameters, in location specific data needs clarification / rectification.

R#21 – We agreed, basic information regarding the data collection of soil and vegetation sampling are now mentioned in detail as per suggestion. (L. 98-110, 112-114, 121-125, 136-138)

Further, the following points / comments need clarification / improvement.

Line no. 24-31: The abstract needs little improvement.

R#22 – Improved.

Line no. 32 & 100-104: The location can be provided with district / provinces / state, etc. in Pakistan.

R#23 – Required details provided in both the sections. (L. 11, 90-91)

Line no. 106: What are the growing seasons targeted? The time period of the study also be specified.

R#24 – We now added the months that represent the growing season in the area, which is from March to October. (L. 97)

Line no. 116-118: What is the deepness of soil sampling? Whether the soil has been collected for each stands / transect, please clarify.

R#25 – “Soil samples of 200-400grams were collected from three randomly selected transects (0-30cm depth) within each sampling stand of the studied vegetation area.” For more details, please see section: Environmental gradients. (L. 121-125)

Line no. 119-121: The duration, intervals, etc. of data obtained from the handheld weather station and position / place of the weather station for the study are not clear.

R#26 – We used a small remote weather station and recorded the data at transect level and then averaged it to stand level. (L. 136-138)

Line no. 177: Details of 244 species recorded may be supplemented.

R#27 – Two new supplementary tables have been with a summary of the environmental variables and the species composition based on the importance value index.

Line no. 182-186: The community may be specified as Trees, Shrubs, Grass, etc. and can be analysed properly. For example, the presence of Indigofera is shrub and Cynodon is grass and the landscape is shrubland or grassland. On the other hand, the presence of Pinus and Cedrus are clearly representing tree communities. Whether the landscape is having patches of vegetation of trees / shrub / grass composition or uniform vegetation dominated by these group, need clarification.

R#28 – As per suggestion, the required details have been mentioned following the communities. For instance, the IHC community was primarily found in the lower mountainous ranges (1932.3-2338.4 m.a.s.l), where the dominating flora was a combination of shrub and herb species owing to the existence of a substantial herbaceous layer of Cynodon dactylon, which carpeted the landscape alongside Indigofera heterantha patches……shrubby associates. (L. 237-246)

Line no. 190-194: Any specific observation on Cedrus, Pinus, etc. Because, we could not find the all vegetation in all aspects, especially the studied tree species. 

R#29 – Please see R#28 for the answer about this question. (L. 237-246, 264-266)

Discussion: Need revision in view of answering the above questions.

R#30 – We provided more details about how the variables can influence plant community. We cordially ask to check the lines (L. 327-329, 367-369, 376-382, 398-432), where we added information about the influence of other variables in the community structure.

The manuscript holds excellent statistical analysis. Although, basic collection data on vegetation assessment and related environmental parameters are missing, and can be addressed.

We cordially appreciate your positive words, however, we have considered and addressed each of your suggestion accordingly.

---

## [Decision Letter · Decision Letter 1]

20 Sep 2021

PONE-D-21-11989R1Environmental variables drive plant species composition and distribution in the moist temperate forests of Northwestern Himalaya, PakistanPLOS ONE

Dear Dr. Rahman,

Thank you for submitting your manuscript to PLOS ONE. After careful consideration, we feel that it has merit but does not fully meet PLOS ONE’s publication criteria as it currently stands. Therefore, we invite you to submit a revised version of the manuscript that addresses the points raised during the review process. All the three reviewers have provided positive recommendations for your MS. There are some minor issues that needs to be resolved. Please look into the Editorial and reviewer comments provided below and revise the MS accordingly. 

We look forward to receiving your revised manuscript.

Kind regards,

Bhoj Kumar Acharya, PhD

Academic Editor

PLOS ONE

Journal Requirements:

**Editor Comments**:

The authors have studied the variation of plant communities along environmental gradients in the Moist North Western Himalaya, and linked the plant communities with various environmental variables. The article is well written with clarity in introduction, methods (including data analysis), results and discussion. In the first round of review, three independent expert reviewers provided valuable comments which were mostly addressed by the authors. The revised version was sent to all the three same previous reviewers and all of them have positively commented on the MS. I have once again thoroughly evaluated the MS and provided some editorial comments. I suggest the authors to look into the comments (as sticky notes in the attached pdf manuscript files) and address them critically. I also suggest the authors to provide some more details in the sampling design and methodology section as pointed by reviewer 1. Once all these minor issues are resolved, the MS may be considered for publication. I request the authors to revise the MS quickly and submit the same. The MS may not be sent for further external review but will be evaluated by the academic editor before rendering the final decision. Looking forward to the revised MS at an early date.

Reviewers' comments:

Reviewer's Responses to Questions

**Comments to the Author**

1. If the authors have adequately addressed your comments raised in a previous round of review and you feel that this manuscript is now acceptable for publication, you may indicate that here to bypass the “Comments to the Author” section, enter your conflict of interest statement in the “Confidential to Editor” section, and submit your "Accept" recommendation.

Reviewer #1: All comments have been addressed

Reviewer #2: All comments have been addressed

Reviewer #3: All comments have been addressed

2. Is the manuscript technically sound, and do the data support the conclusions?

Reviewer #1: Yes

Reviewer #2: Yes

Reviewer #3: Yes

3. Has the statistical analysis been performed appropriately and rigorously? 

Reviewer #1: Yes

Reviewer #2: Yes

Reviewer #3: Yes

4. Have the authors made all data underlying the findings in their manuscript fully available?

Reviewer #1: Yes

Reviewer #2: Yes

Reviewer #3: Yes

5. Is the manuscript presented in an intelligible fashion and written in standard English?

Reviewer #1: Yes

Reviewer #2: Yes

Reviewer #3: Yes

6. Review Comments to the Author

Reviewer #1: The authors addressed the most of the issues specified.

Although, some more details on basic collection data on vegetation assessment and related environmental parameters are still missing in the manuscript. The similar observations also was observed by other reviewers also. So, I request the authors to kindly some more details on data collected on field, especially ecological and environmental data.

Reviewer #2: (No Response)

Reviewer #3: The revised manuscript draft seems much improved and informative to its initial draft. I believe all the suggestions have been incorporated to the manuscript and now it has merit to be accepted for the publication.

With Best Regards

7. PLOS authors have the option to publish the peer review history of their article (what does this mean?). If published, this will include your full peer review and any attached files.

Reviewer #1: **Yes: **K Chandra Sekar

Reviewer #2: **Yes: **Shweta Basnett

Reviewer #3: **Yes: **Dr. Aseesh Pandey

---

## [Author Response · Author response to Decision Letter 1]

12 Nov 2021

Revision notes

Reviewers and Editor’s comments; author responses.

Reviewer(s)’ Comments to Author:

Editor’s Comments to Author:

Comment: The authors have studied the variation of plant communities along environmental gradients in the Moist North Western Himalaya, and linked the plant communities with various environmental variables. The article is well written with clarity in introduction, methods (including data analysis), results and discussion. In the first round of review, three independent expert reviewers provided valuable comments which were mostly addressed by the authors. The revised version was sent to all the three same previous reviewers and all of them have positively commented on the MS. I have once again thoroughly evaluated the MS and provided some editorial comments. I suggest the authors to look into the comments (as sticky notes in the attached pdf manuscript files) and address them critically. I also suggest the authors to provide some more details in the sampling design and methodology section as pointed by reviewer 1. Once all these minor issues are resolved, the MS may be considered for publication. I request the authors to revise the MS quickly and submit the same. The MS may not be sent for further external review but will be evaluated by the academic editor before rendering the final decision. Looking forward to the revised MS at an early date.

Response: We really appreciate the reviewers for very constructive criticism and acceptance of our work. We are thankful to the Editor for highlighting such important points, which made the revised version much better. We have considered all the raised points and corrected the manuscript accordingly.

Comment: The Editor highlighted suggestions in the pdf file.

Response: We are thankful to the Editor for highlighting such important points, which made the revised version much better. We have considered all the raised points and corrected the manuscript accordingly. Each Table and Figure legend/caption were corrected with proper and complete details. The analysis and the linked results that were highlighted to be deleted have been done as suggested. We cordially ask you to check the track version for corrections.

Comment: Sorenson dissimilarly index (βsor) is an incidence based index which can be partitioned into turnover (βsim) and nestedness (βnes) components. Simpson is a different index. Again why authors have used only incidence based index? Why not abundance based index because abundance-based β-diversity can be estimated as Bray-Curtis dissimilarity index (dBC) and then partitioned into balanced variation (dBC-bal) and abundance gradient components (dBC-gra). Since authors have abundance data, both the approaches could give better result. Please see the following article for details:

Sharma, K., B.K. Acharya, G. Sharma, D. Valente, M.R Pasimeni, I. Petrosillo, and T. Selvan. 2020. Land use effect on butterfly alpha and beta diversity in the Eastern Himalaya, India. Ecological Indicators 110: 105605.

Response: We agree that when working with abundance, abundance-based B-diversity estimated as dBC is the best option. However, in all our analyses we used the importance value (IV), as described in the Vegetation Sampling and plant identification subtopic in Material and Methods, to standardize our analyses over the manuscript. IV takes in account relative frequency, relative density, and relative dominance. In this way, we chose to use the Bsor index based on the incidence to decrease potential confounding effects of using IV when calculating dBC. In few words, we used a more conservative Beta diversity analysis. For these reasons, we cordially ask you to keep the incidence-based Beta diversity analysis in our study.

---

## [Editor Report · Decision Letter 2]

16 Nov 2021

Environmental variables drive plant species composition and distribution in the moist temperate forests of Northwestern Himalaya, Pakistan

PONE-D-21-11989R2

Dear Dr. Rahman,

We’re pleased to inform you that your manuscript has been judged scientifically suitable for publication and will be formally accepted for publication once it meets all outstanding technical requirements.

Kind regards,

Bhoj Kumar Acharya, PhD

Academic Editor

PLOS ONE

---

## [Editor Report · Acceptance letter]

14 Feb 2022

PONE-D-21-11989R2 

Environmental variables drive plant species composition and distribution in the moist temperate forests of Northwestern Himalaya, Pakistan 

Dear Dr. Rahman:

I'm pleased to inform you that your manuscript has been deemed suitable for publication in PLOS ONE. Congratulations! Your manuscript is now with our production department. 

Kind regards, 

on behalf of

Dr. Bhoj Kumar Acharya 

Academic Editor

PLOS ONE